# Routing by Reaching: Composition of Pre-trained GFlowNets for Multi-Objective Generation

**Seokwon Yoon** [1]  **Youngbin Choi** [2]  **Seunghyuk Cho** [2]  **Seungbeom Lee** [2]  **MoonJeong Park** [2]  **Dongwoo Kim** [1 2]

## Abstract

Generative Flow Networks (GFlowNets) learn to sample diverse candidates in proportion to a reward function, making them well-suited for scientific discovery, where exploring multiple promising solutions is crucial. Further extending GFlowNets to multi-objective settings has attracted growing interest as real-world applications often involve multiple, conflicting objectives. However, existing approaches require joint training for each combination of objectives, meaning that any change in the objective set necessitates retraining from scratch. We propose a framework that composes pre-trained GFlowNets at inference time, enabling rapid adaptation without fine-tuning or retraining. Importantly, our framework is flexible, capable of handling diverse reward combinations ranging from linear scalarization to complex nonlinear operators, which are often handled separately in previous literature. We prove that our method exactly recovers the target distribution for linear scalarization, and quantify the approximation quality for nonlinear operators through a distortion factor. Experiments on a synthetic 2D grid and real-world molecule generation tasks demonstrate that our approach achieves performance comparable to baselines. The code is available at https://github.com/ml-postech/gflownet-composition.

## 1. Introduction

Generative models for structured discrete objects, e.g., molecules (Jin et al., 2020; Guan et al., 2023; Huang et al., 2024; Qu et al., 2024), graphs (Liao et al., 2019; Jo et al., 2022; Martinkus et al., 2022; Jang et al., 2024; Zhou et al., 2024), and biological sequences (Ingraham et al., 2019; Truong Jr. & Bepler, 2023; Koh et al., 2025), have become essential tools for scientific discovery. Among these, Generative Flow Networks (GFlowNets) (Bengio et al., 2021; 2023; Jain et al., 2023a) have emerged as a particularly compelling framework for sampling diverse candidates from distributions proportional to a given reward function. Unlike traditional optimization methods (Sutton & Barto, 1998; Schulman et al., 2015; 2017) that converge to a single high-reward solution, GFlowNets explore multiple promising candidates rather than concentrating on a single optimum.

While standard GFlowNets are trained with respect to a single reward function, many real-world scientific applications require balancing multiple, often conflicting objectives simultaneously (Nouhi et al., 2022; Harris et al., 2023; Lee et al., 2025; Ran et al., 2025). In practice, the optimal trade-off among these objectives is rarely known a priori, requiring practitioners to flexibly explore diverse compositions of objectives to identify candidates that best suit their downstream requirements. This has motivated recent efforts to extend GFlowNets to multi-objective settings through two primary strategies: scalarization and logical operators. Scalarization approaches aggregate multiple objectives into a single scalar reward via a weighted sum to sample candidates across diverse trade-off profiles (Jain et al., 2023b; Zhu et al., 2023). Alternatively, logical operators, as exemplified by compositional sculpting (Garipov et al., 2023), combine the distributions induced by separately trained GFlowNets through a classifier-guided framework, supporting operators such as the harmonic mean for conjunction and the contrast for subtraction.

Despite these approaches enabling multi-objective generation, they suffer from two critical limitations. First, both strategies require joint training for each combination of objectives, limiting their applicability to a predefined set of reward functions defined during the training phase. For instance, preference-conditioned GFlowNets (Jain et al., 2023b; Zhu et al., 2023) must be retrained from scratch to incorporate new objectives, whereas compositional sculpting requires training a dedicated auxiliary classifier for each new set of objectives. Both approaches incur substantial

[1]Department of Computer Science & Engineering, POSTECH, South Korea [2]Graduate School of Artificial Intelligence, POSTECH, South Korea. Correspondence to: Dongwoo Kim <dongwoo.kim@postech.ac.kr>.

*Proceedings of the 43rd International Conference on Machine Learning*, Seoul, South Korea. PMLR 306, 2026. Copyright 2026 by the author(s).

overhead that often exceeds the cost of training a single-objective GFlowNet (Garipov et al., 2023). Second, existing methods typically address only a single formulation of multi-objective combination, either scalarization or logical composition, lacking a unified framework for both. Consequently, developing a methodology that directly composes pre-trained GFlowNets across arbitrary reward sets and diverse composition schemes remains a significant open challenge.

In this work, we present a framework for composing pre-trained GFlowNets by mixing their forward policies at inference time. Our key insight is that the likelihood of visiting a state in each model provides natural weights for combining their generation processes. Intuitively, this prioritizes the model that considers the current state more relevant to its objective. We provide a theoretical analysis of our mixing rule for linear scalarization, which shows that the mixing policy exactly induces the target distribution. For general nonlinear composition operators, we analyze the $L_1$ distance between the induced and target distributions, and empirically show that our mixing rule provides an accurate approximation in high-density regions where it matters most.

We conduct experiments on a synthetic 2D grid domain and real-world molecule generation tasks, including both fragment- and atom-based settings. On the synthetic 2D grid domain, our mixing policy achieves substantially lower error than preference-conditioned baselines for linear scalarization, maintaining stable performance even as the number of objectives increases. For logical operators, our method performs comparably to classifier-guided mixing while requiring no additional training. On molecular generation, our approach matches or exceeds the sample quality of baselines that require joint training for each objective combination, and for logical operators, our method is significantly faster than classifier-guided composition during inference. Overall, experimental results demonstrate that our method enables diverse compositions without additional training at the composition stage while incurring minimal inference overhead.

## 2. Related Work

**Multi-objective generation in GFlowNets.** Two primary strategies have been used to extend GFlowNets to multi-objective settings: scalarization and logical operators. Scalarization approaches transform multiple objectives into a single scalar reward through weighted combinations. Jain et al. (2023b) proposes MOGFN, which trains a single GFlowNet conditioned on preference vectors, enabling sampling across different trade-offs without retraining for each preference. Zhu et al. (2023) extends this idea with hypernetwork-based GFlowNets (HN-GFN), and Chen & Mauch (2024) introduces order-preserving GFlowNets that

learn from a partial order over candidates without explicit reward scalarization. While these methods effectively explore the Pareto front, they require training a new model whenever a new objective is introduced, preventing the reuse of the existing model.

Logical operators offer an alternative by combining distributions learned by separately trained GFlowNets. Garipov et al. (2023) proposes compositional sculpting, which introduces logical operators such as the harmonic mean for conjunction and the contrast for subtraction. However, their method requires training an auxiliary classifier for each set of objectives, which often exceeds the cost of training the base GFlowNet itself.

Our work provides a unified framework that covers both directions. By mixing pre-trained GFlowNet forward policies at inference time, we support scalarization across varying preference trade-offs and logical operators such as conjunction and subtraction.

**Model composition.** Building complex distributions by composing pre-trained models has a rich history in machine learning. Products of experts (Hinton, 1999; 2002) combine multiple probabilistic models by multiplying their densities. In energy-based models, composition enables constructing complex concepts through algebraic operations on energy functions (Du et al., 2020; 2021). Recent work extends these ideas to diffusion models, enabling compositional generation through score function combination (Liu et al., 2022; Du et al., 2023). Classifier guidance (Dhariwal & Nichol, 2021; Ho & Salimans, 2021) further provides post-hoc control by steering generation toward desired attributes without retraining the base model. Beyond probability-space operations, weight-space averaging provides an alternative approach: model soups (Wortsman et al., 2022) and rewarded soups (Ramé et al., 2023) combine fine-tuned models by interpolating their parameters. For GFlowNets, compositional sculpting (Garipov et al., 2023) represents the first attempt at model composition, but requires training a classifier for each new composition. Our work proposes training-free model composition for multi-objective generation.

## 3. Background

In this section, we review the foundations of GFlowNets, including the reaching probability. We also describe the multi-objective generation problem.

### 3.1. Generative Flow Networks (GFlowNets)

GFlowNets (Bengio et al., 2021; 2023) are generative models designed to sample structured objects $x$, such as graphs or molecules, from a discrete space $\mathcal{X}$. The primary goal is to sample objects from a distribution $p(x)$ proportional to a

given non-negative reward function $R(x)$:

$$p(x) = \frac{R(x)}{Z}, \quad \text{where } Z = \sum_{x \in \mathcal{X}} R(x).$$

In practice, a temperature parameter $\beta$ is often used to control the sharpness of the distribution, yielding $p(x) \propto R(x)^\beta$. Increasing $\beta$ concentrates probability mass on high-reward candidates (Bengio et al., 2021).

To sample from such a distribution, GFlowNets model the generation as a sequential construction process governed by a stochastic forward policy $p_F$. Starting from an initial state $s_0$, the policy samples an action to transition to a new state, progressively building up the object until reaching a terminating state $x \in \mathcal{X}$ and transitioning to the sink state $s_f$, which marks the completion of a trajectory. Formally, this process is defined over a directed acyclic graph (DAG) $(\mathcal{S}, \mathcal{A})$, where $\mathcal{S}$ is a set of states and $\mathcal{A} \subseteq \mathcal{S} \times \mathcal{S}$ is a set of directed edges representing actions. There exist two distinguished states: a unique initial state $s_0 \in \mathcal{S}$ where all trajectories begin, and a unique sink state $s_f \in \mathcal{S}$ where all trajectories terminate. The set of terminating states $\mathcal{X} \subset \mathcal{S}$ consists of states that can transition to $s_f$ via a terminating edge, though they may also transition to other states in $\mathcal{S}$. The forward policy $p_F$ defines a distribution $p_F(\cdot|s)$ over the children of state $s \in \mathcal{S} \backslash \{s_f\}$, inducing a distribution over complete trajectories $\tau = (s_0 \to s_1 \to \cdots \to s_n = x \to s_f)$, where $x \in \mathcal{X}$ represents the constructed object.

**Reaching probability.** A key quantity that will play a central role in our method is the reaching probability $u(s)$, defined as the probability of visiting state $s$ when starting from $s_0$ under the forward policy $p_F$ (Bengio et al., 2023):

$$u(s) := \sum_{\tau \in \mathcal{T}_{s_0,s}} \prod_{t=1}^{|\tau|} p_F(s_t \mid s_{t-1}), \tag{1}$$

where $\mathcal{T}_{s_0,s}$ denotes the set of all trajectories from $s_0$ to $s$, and $|\tau|$ denotes the length of $\tau$. By definition, $u(s_0) = 1$ since every trajectory begins at $s_0$. Intuitively, $u(s)$ captures how much probability mass the policy routes through state $s$, and satisfies the recursion:

$$u(s) = \sum_{s_*:(s_* \to s) \in \mathcal{A}} u(s_*) \, p_F(s \mid s_*). \tag{2}$$

For a terminating state $x \in \mathcal{X}$, the terminating state probability decomposes as:

$$p(x) = u(x) \cdot p_F(s_f \mid x). \tag{3}$$

This factorization separates the probability of reaching $x$ from the probability of stopping at $x$.

### 3.2. Multi-Objective Generation in GFlowNets

Prior work in GFlowNets has addressed the multi-objective generation problem by enabling a single model to cover a family of compositions over multiple reward functions $\mathbf{R}(x) = (R_1(x), \ldots, R_k(x))$. Jain et al. (2023b); Zhu et al. (2023) focus on scalarization, where multiple objectives are combined via a weighted sum, scaled by temperature $\beta$: $\tilde{R}(x) = (\sum_{i=1}^k \omega_i R_i(x))^\beta$. By conditioning the policy on the preference vector $\boldsymbol{\omega} := (\omega_1, \cdots, \omega_k)$, a single model learns to sample from distributions corresponding to arbitrary weight combinations within a fixed set of reward functions.

Compositional sculpting (Garipov et al., 2023) takes a different approach: given separately trained GFlowNets for each reward function, it enables sampling from distributions defined by logical operators (e.g., conjunction, subtraction) over the component distributions through training an auxiliary model. Specifically, it introduces two logical operators. The harmonic mean operator ($\otimes$) acts as a conjunction, assigning high probability only where all component distributions have high density: $(p_1 \otimes p_2)(x) \propto \frac{p_1(x)p_2(x)}{p_1(x)+p_2(x)}$. The contrast operator ($\circleddash$) functions as a subtraction, highlighting regions where the first distribution dominates: $(p_1 \circleddash p_2)(x) \propto \frac{p_1(x)^2}{p_1(x)+p_2(x)}$. These binary operators can be extended to more than two distributions through chaining. For a detailed discussion of the generalization of these operators, please refer to Garipov et al. (2023).

Following the setting of compositional sculpting, we assume access to a set of pre-trained GFlowNets $\{p_{i,F}\}_{i=1}^k$, where each $p_{i,F}$ is trained to sample from a distribution proportional to its corresponding reward function $R_i(x)$.

## 4. Method

In this section, we propose a framework that composes pre-trained GFlowNets by mixing their forward policies at inference time (Section 4.1). We prove its exactness for linear scalarization and analyze the approximation error for nonlinear operators (Section 4.2).

### 4.1. Mixture of GFlowNets

As described in Section 3.2, we assume access to $k$ pre-trained GFlowNets, where the $i$-th GFlowNet is trained on reward function $R_i(x)$ and induces a terminating state distribution $p_i(x) \approx R_i(x)/Z_i$. Our goal is to sample from a target distribution $p_M^*(x) \propto \mathcal{G}(p_1(x), \ldots, p_k(x))$, where $\mathcal{G}$ specifies how the component distributions are combined. Crucially, we aim to achieve this using only the pre-trained forward policies at inference time, without any additional training.

For the $i$-th component GFlowNet, we denote its forward

policy by $p_{i,F}$, reaching probability by $u_i$, and terminating state distribution by $p_i$. We define the mixing policy as:

$$p_{M,F}(s'|s) = \frac{\mathcal{G}\big(u_1(s)p_{1,F}(s'|s),\ldots,u_k(s)p_{k,F}(s'|s)\big)}{N_M(s)},$$
(4)

where the local normalization constant $N_M(s)$ is

$$\sum_{s':(s\to s')\in\mathcal{A}} \mathcal{G}\big(u_1(s)p_{1,F}(s'|s),\ldots,u_k(s)p_{k,F}(s'|s)\big).$$
(5)

The mixing policy weights each model's transition probability $p_{i,F}(s'|s)$ by its reaching probability $u_i(s)$, then combines them through the composition function $\mathcal{G}$.

**Computing the reaching probability.** Computing $u_i(s)$ directly via the recursion Eq. (2) requires summing over all parent states, which is intractable in large state spaces. Instead, we use the identity $u_i(s) = F_i(s)/Z_i$ (see Appendix A.2.1 for derivation), where $F_i(s)$ is the state flow and $Z_i = F_i(s_0)$ is the total flow. The problem thus reduces to obtaining $F_i(s)$. Depending on the training objective, the state flow may be modeled explicitly or only implicitly through the learned policy. We therefore provide two routes for obtaining $F_i(s)$, covering both cases:

- **Model $F$.** When the GFlowNet is trained with flow matching (Bengio et al., 2021), detailed balance (Bengio et al., 2023), or sub-trajectory balance (Madan et al., 2023), $F_i(s)$ is explicitly parameterized and can be read directly from the learned model.

- **DB $F$.** For objectives that do not parameterize $F_i(s)$ (e.g., trajectory balance (Malkin et al., 2022)), we recover $F_i(s)$ via the detailed balance condition $F_i(s')\,p_{i,B}(s|s') = F_i(s)\,p_{i,F}(s'|s)$, which holds at convergence for any GFlowNet. Unrolling this identity along a trajectory gives $F_i(s_t) = Z_i \prod_{j=1}^{t} p_{i,F}(s_j|s_{j-1})/p_{i,B}(s_{j-1}|s_j)$, which can be accumulated on the fly during trajectory generation (see Appendix A.2.3 for the full derivation).

### 4.2. Analysis of the Induced Distribution

To understand when our mixing policy recovers the target distribution, we first derive its general form. The distribution induced by our mixing policy takes the form:

$$\begin{aligned} p_M(x) &= u_M(x) \cdot p_{M,F}(s_f \mid x) \\ &= \frac{u_M(x)}{N_M(x)} \cdot \mathcal{G}\big(p_1(x),\ldots,p_k(x)\big), \end{aligned}$$
(6)

where the *distortion factor* $\delta(x) := u_M(x)/N_M(x)$ measures how much the induced distribution deviates from the target. When $\delta(x)$ is constant across all $x$, our mixing policy exactly recovers the target distribution $p_M^*(x) \propto$

$\mathcal{G}(p_1(x),\ldots,p_k(x))$. We examine linear scalarization, where exact recovery is guaranteed, and nonlinear operators, where approximations may arise.

**Linear scalarization.** Linear scalarization combines multiple objectives via a weighted sum. Given a preference vector $\boldsymbol{\omega} = (\omega_1,\ldots,\omega_k)$ with $\omega_i \geq 0$, the target distribution is:

$$p_M^*(x) \propto \sum_{i=1}^{k} \omega_i R_i(x).$$
(7)

The corresponding composition function is $\mathcal{G}(p_1,\ldots,p_k) = \sum_{i=1}^{k} \omega_i Z_i p_i$. For this setting, $\delta(x)$ becomes constant, yielding exact recovery:

**Proposition 4.1** (Exact realization for linear scalarization). *Given $k$ GFlowNets with terminating distributions $p_i(x) \propto R_i(x)$, the mixing policy Eq. (4) under linear scalarization exactly realizes the target distribution:*

$$p_M(x) = p_M^*(x) \propto \sum_{i=1}^{k} \omega_i R_i(x),$$
(8)

*where $p_M$ is the distribution induced by our mixing policy.*

The proof is provided in Appendix A.1. Under linear scalarization, this provides a method for multi-objective composition of GFlowNets with provable exactness guarantees.

**Nonlinear composition operators.** For nonlinear composition operators, including scalarization with $\beta \neq 1$ where $\mathcal{G}(p_1,\ldots,p_k) = (\sum_{i=1}^{k} \omega_i Z_i p_i)^\beta$, and the logical operators introduced in Section 3.2, $\delta(x)$ is not guaranteed to be constant, potentially introducing deviations from the target distribution. We empirically analyze in Section 5.3 that $\delta(x)$ remains approximately constant in high-reward regions most relevant to sampling, suggesting that our mixing policy provides accurate approximations where it matters most.

## 5. Synthetic Experiments

We conduct synthetic experiments on a 2D grid domain to validate our framework under controlled conditions where ground-truth distributions are computable.

### 5.1. Experimental Settings

**Tasks.** We use a $32 \times 32$ 2D grid domain (Bengio et al., 2021), where the policy learns to navigate from the start state to high-reward regions. In this setting, each state corresponds to a grid cell, and the start state is the upper-left cell $s_0 = (0, 0)$. At each state, the policy takes one of three actions: move right, move down, or terminate at the current state. We utilize five standard benchmark rewards and three synthetic circle rewards, all defined over the grid cells and normalized to the range $[0, 1]$ (Fig. A1).

*Table 1.* $L_1$ error on linear scalarization in a 2D grid with a varying number of objectives. $N$ Obj. denotes scalarization with $N$ objectives, where the specific objectives used are detailed in Table A1. Results are averaged over 128 evenly spaced preference vectors.

|          | 2 Obj. | 3 Obj. | 4 Obj. | 5 Obj. |
|----------|--------|--------|--------|--------|
| MOGFN    | 0.021  | 0.027  | 0.042  | 0.048  |
| HN-GFN   | 0.017  | 0.021  | 0.032  | 0.035  |
| Ensemble | 0.117  | 0.098  | 0.113  | 0.111  |
| Ours     | **0.003** | **0.003** | **0.003** | **0.003** |

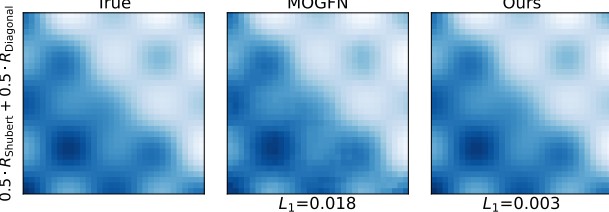

*Figure 1.* Qualitative result of scalarization on a 2D grid domain. We visualize the density over the grid for the true distribution (left), MOGFN (middle), and ours (right).

We conduct experiments on both scalarization and logical operators. For scalarization, we select subsets of size 2 to 5 from the five base rewards, use 128 evenly spaced preference vectors $\omega$ on the simplex for each subset, and evaluate the mixing policy under each preference. For logical operators, we evaluate various combinations of harmonic mean and contrast operations applied to the base and circle rewards.

**Baselines.** For scalarization, we consider preference-conditioned GFlowNets, including MOGFN (Jain et al., 2023b) and HN-GFN (Zhu et al., 2023). For logical operators, we compare against the classifier guidance approach of Garipov et al. (2023). For both settings, we also include an ensemble baseline that mixes forward policies without the reaching probability $u_i(\cdot)$ to validate the importance of flow-based weighting. Our mixing policy uses the Model $F$ route (Section A.2.2) to compute reaching probabilities.

**Evaluation metrics.** In this setting, where the true target distribution and induced target distribution can be computed exactly, we evaluate the approximation quality using the $L_1$ error, i.e., $\sum_x |p_{\text{model}}(x) - p_{\text{target}}(x)|$, between the distribution induced by our mixing policy and the ground-truth target distribution, following Bengio et al. (2021).

We set $\beta = 1$ in this task. Further details on experimental settings and additional results are provided in Appendix B.1.

### 5.2. Results

**Scalarization.** Table 1 presents the average $L_1$ error over 128 evenly spaced preference vectors for varying numbers of

*Table 2.* $L_1$ error on logical operators in a 2D grid. We evaluate harmonic mean ($\otimes$) and contrast ($\circ$) operators across different reward pairs. Additional results are provided in Table A2.

|   | Classifier guidance | Ensemble | Ours |
|---|--------------------|----------|------|
| *Harmonic mean ($\otimes$)* | | | |
| $p_{\text{Shubert}} \otimes p_{\text{Sphere}}$ | 0.158 | 0.145 | **0.136** |
| $p_{\text{Shubert}} \otimes p_{\text{Diagonal}}$ | **0.142** | 0.190 | 0.180 |
| $p_{\text{Circle1}} \otimes p_{\text{Circle2}}$ | 0.397 | 0.453 | **0.229** |
| *Contrast ($\circ$)* | | | |
| $p_{\text{Shubert}} \circ p_{\text{Sphere}}$ | 0.175 | 0.175 | **0.111** |
| $p_{\text{Sphere}} \circ p_{\text{Shubert}}$ | 0.190 | 0.162 | **0.116** |
| $p_{\text{Shubert}} \circ p_{\text{Diagonal}}$ | 0.150 | 0.190 | **0.106** |
| $p_{\text{Diagonal}} \circ p_{\text{Shubert}}$ | **0.153** | 0.190 | 0.158 |
| $p_{\text{Circle1}} \circ p_{\text{Circle2}}$ | 0.245 | 0.356 | **0.231** |
| $p_{\text{Circle2}} \circ p_{\text{Circle1}}$ | 0.241 | 0.390 | **0.070** |

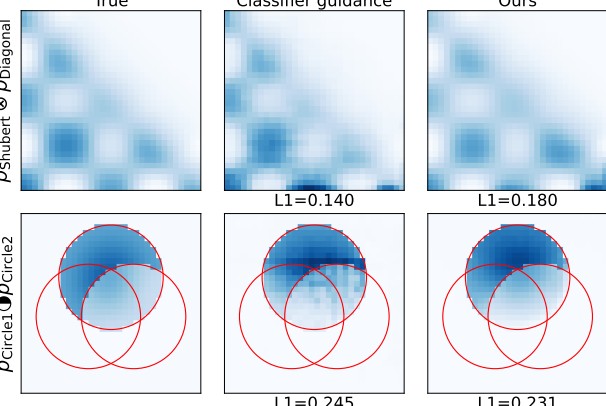

*Figure 2.* Qualitative result of logical operations on a 2D grid domain. We visualize the density over the grid for the true distribution (left), classifier guidance (middle), and ours (right).

objectives. Our mixing policy consistently outperforms the baselines across all settings. Notably, as the number of objectives increases from 2 to 5, our method maintains nearly constant error, whereas MOGFN and HN-GFN degrade with more objectives. The ensemble exhibits substantially higher $L_1$ error than ours, demonstrating that naive mixing without reaching probability weighting fails to approximate the target distribution.

For the two-objective setting, we additionally train single-objective GFlowNets separately for each of five fixed preference vectors. As shown in Fig. A2, our mixing policy outperforms even these individually trained models. Fig. 1 shows the qualitative results. Our mixing policy more accurately recovers the multi-modal structure of the true distribution, whereas MOGFN produces slightly blurred modes.

**Logical operators.** Table 2 compares the ground-truth composition distributions with those induced by our mixing rule and baselines. Our mixing policy performs compara-

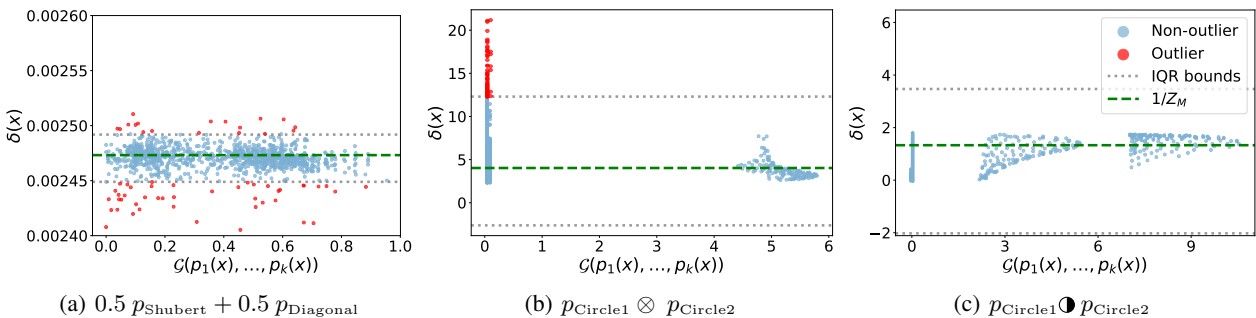

*Figure 3.* Distortion factor $\delta(x) = u_M(x)/N_M(x)$ vs. unnormalized target density $\mathcal{G}(p_1(x), \ldots, p_k(x))$ on the 2D grid domain. Red points indicate outliers, defined as states where $\delta(x)$ falls outside $[Q_1 - 1.5 \cdot \mathrm{IQR}, Q_3 + 1.5 \cdot \mathrm{IQR}]$ (gray dotted lines), where $Q_1, Q_3$ are the 25th/75th percentiles and $\mathrm{IQR} = Q_3 - Q_1$. The green dashed line marks $1/Z_M$ with $Z_M = \sum_x \mathcal{G}(p_1(x), \ldots, p_k(x))$. Additional results are in Fig. A6.

bly to the classifier-guided approach, with results varying across composition types: our method outperforms on some compositions while the classifier-guided baseline performs better on others. Notably, our method achieves this without any additional training, whereas the classifier-guided approach requires training a new auxiliary classifier for each set of objectives. Our mixing policy also improves over the ensemble baseline across all settings, indicating that the reaching probability $u_i(\cdot)$ provides a more refined weighting than uniform mixing of forward policies.

Fig. 2 shows qualitative results of classifier-guided and our mixing policy. Both successfully achieve the intended behavior of each operator: concentrating probability mass on regions where both component distributions have high density for the harmonic mean, and isolating regions where only the first distribution dominates for contrast.

### 5.3. Analysis of Approximation Quality

As discussed in Section 4.2, for nonlinear composition operators, the distortion factor $\delta(x)$ is not guaranteed to be constant, potentially introducing deviations from the target distribution. Here, we empirically analyze how $\delta(x)$ behaves in practice.

To quantify the approximation error, we consider the $L_1$ distance between the induced distribution $p_M(x)$ and the target distribution $p_M^*(x)$:

$$L_1 = \sum_x |p_M(x) - p_M^*(x)|$$

$$= \sum_x \left| \delta(x) - \frac{1}{Z_M} \right| \mathcal{G}(p_1(x), \ldots, p_k(x)), \quad (9)$$

where $Z_M = \sum_x \mathcal{G}(p_1(x), \ldots, p_k(x))$. We provide the detailed derivation in Section A.3. To analyze the approximation quality, we compute $1/Z_M$, the distortion factor $\delta(x)$ and the composition value $\mathcal{G}(p_1(x), \ldots, p_k(x))$ for each $x$.

Fig. 3 visualizes how $\delta(x)$ varies with $\mathcal{G}(p_1(x), p_2(x))$ across different composition operators. For linear scalarization, Fig. 3(a) confirms that $\delta(x)$ remains constant at $1/Z_M$ across all samples, as expected from Proposition 4.1. For logical operators, Figs. 3(b) and 3(c) show that $\delta(x)$ is not exactly constant but remains close to $1/Z_M$ in high-composition value regions where $\mathcal{G}(\cdot)$ is large. Deviations primarily occur in low-composition value regions where $\mathcal{G}(\cdot)$ is small, contributing minimally to the overall $L_1$ error. This confirms that our mixing policy provides accurate approximations for samples that contribute most to the target distribution. Additional results are provided in Fig. A6 and Fig. A7.

## 6. Real-world Experiments

We evaluate our framework on real-world molecular generation tasks to demonstrate its practical applicability.

### 6.1. Experimental Settings

**Tasks.** We evaluate our framework on two tasks: fragment- and atom-based molecule generation. In fragment-based generation, molecules are assembled from a pre-defined vocabulary of 72 fragments. At each step, the policy selects a fragment and specifies attachment points on both the existing structure and the new fragment, with a maximum of 9 fragments per molecule. We employ three reward functions: (i) SEH, which measures the predicted binding affinity to soluble epoxide hydrolase using a pre-trained proxy model (Bengio et al., 2021); (ii) SA, which estimates synthetic accessibility (Ertl & Schuffenhauer, 2009); and (iii) QED, which quantifies drug-likeness (Bickerton et al., 2012). All rewards are normalized to $[0, 1]$ with larger values indicating better molecules.

For atom-based generation (QM9), molecules are constructed atom-by-atom and bond-by-bond. Each state is rep-

*Table 3.* Results of scalarization on molecule generation tasks. We report the average reward and diversity of the top-10 samples across different objective combinations (mean $\pm$ std over 3 seeds). *ALL* denotes all three objectives (SEH/GAP, SA, and QED).

| | | Reward $\uparrow$ | | | | Diversity $\uparrow$ | | | |
|---|---|---|---|---|---|---|---|---|---|
| | | MOGFN | HN-GFN | Ours (Model $F$) | Ours (DB $F$) | MOGFN | HN-GFN | Ours (Model $F$) | Ours (DB $F$) |
| **Fragment** | SEH-SA | $0.838_{\pm 0.002}$ | $\mathbf{0.839}_{\pm 0.005}$ | $0.836_{\pm 0.001}$ | $0.835_{\pm 0.003}$ | $\mathbf{0.573}_{\pm 0.001}$ | $\mathbf{0.573}_{\pm 0.001}$ | $0.562_{\pm 0.005}$ | $0.554_{\pm 0.006}$ |
| | SEH-QED | $0.764_{\pm 0.006}$ | $0.711_{\pm 0.012}$ | $0.775_{\pm 0.010}$ | $\mathbf{0.777}_{\pm 0.006}$ | $\mathbf{0.571}_{\pm 0.007}$ | $0.568_{\pm 0.007}$ | $0.564_{\pm 0.006}$ | $0.565_{\pm 0.013}$ |
| | SA-QED | $0.783_{\pm 0.006}$ | $0.790_{\pm 0.009}$ | $\mathbf{0.797}_{\pm 0.010}$ | $0.790_{\pm 0.008}$ | $0.630_{\pm 0.008}$ | $\mathbf{0.650}_{\pm 0.006}$ | $0.624_{\pm 0.007}$ | $0.621_{\pm 0.012}$ |
| | ALL | $0.723_{\pm 0.004}$ | $0.670_{\pm 0.001}$ | $0.741_{\pm 0.009}$ | $\mathbf{0.742}_{\pm 0.010}$ | $0.608_{\pm 0.003}$ | $0.608_{\pm 0.008}$ | $0.608_{\pm 0.009}$ | $\mathbf{0.610}_{\pm 0.008}$ |
| **QM9** | GAP-SA | $0.799_{\pm 0.022}$ | $0.742_{\pm 0.035}$ | $\mathbf{0.873}_{\pm 0.003}$ | $0.868_{\pm 0.003}$ | $\mathbf{0.919}_{\pm 0.008}$ | $0.912_{\pm 0.003}$ | $0.899_{\pm 0.010}$ | $0.902_{\pm 0.009}$ |
| | GAP-QED | $\mathbf{0.784}_{\pm 0.004}$ | $0.773_{\pm 0.009}$ | $0.778_{\pm 0.004}$ | $0.781_{\pm 0.005}$ | $0.859_{\pm 0.021}$ | $0.865_{\pm 0.019}$ | $0.880_{\pm 0.014}$ | $\mathbf{0.881}_{\pm 0.014}$ |
| | SA-QED | $0.667_{\pm 0.023}$ | $0.707_{\pm 0.001}$ | $\mathbf{0.710}_{\pm 0.013}$ | $0.689_{\pm 0.014}$ | $\mathbf{0.853}_{\pm 0.037}$ | $0.808_{\pm 0.009}$ | $0.778_{\pm 0.040}$ | $0.826_{\pm 0.019}$ |
| | ALL | $0.688_{\pm 0.028}$ | $0.663_{\pm 0.005}$ | $\mathbf{0.727}_{\pm 0.005}$ | $0.719_{\pm 0.004}$ | $\mathbf{0.895}_{\pm 0.007}$ | $0.875_{\pm 0.003}$ | $0.872_{\pm 0.014}$ | $0.882_{\pm 0.010}$ |

*Table 4.* Hypervolume on molecule generation tasks. We report mean $\pm$ std over 3 seeds. *ALL* denotes all three objectives (SEH/GAP, SA, and QED).

| | | Hypervolume $\uparrow$ | | | | |
|---|---|---|---|---|---|---|
| | | MOGFN | HN-GFN | OP-GFN | Ours(Model $F$) | Ours(DB $F$) |
| **Fragment** | SEH-SA | $0.832_{\pm 0.012}$ | $\mathbf{0.842}_{\pm 0.001}$ | $0.807_{\pm 0.026}$ | $0.834_{\pm 0.003}$ | $0.824_{\pm 0.005}$ |
| | SEH-QED | $0.758_{\pm 0.023}$ | $0.747_{\pm 0.015}$ | $0.581_{\pm 0.029}$ | $\mathbf{0.768}_{\pm 0.018}$ | $0.754_{\pm 0.026}$ |
| | SA-QED | $0.781_{\pm 0.014}$ | $0.790_{\pm 0.017}$ | $0.655_{\pm 0.021}$ | $\mathbf{0.792}_{\pm 0.001}$ | $0.787_{\pm 0.018}$ |
| | ALL | $0.665_{\pm 0.012}$ | $0.651_{\pm 0.005}$ | $0.476_{\pm 0.038}$ | $0.674_{\pm 0.021}$ | $\mathbf{0.679}_{\pm 0.023}$ |
| **QM9** | GAP-SA | $0.976_{\pm 0.051}$ | $0.958_{\pm 0.089}$ | $0.972_{\pm 0.062}$ | $\mathbf{1.000}_{\pm 0.008}$ | $0.981_{\pm 0.015}$ |
| | GAP-QED | $0.642_{\pm 0.004}$ | $\mathbf{0.665}_{\pm 0.009}$ | $\mathbf{0.665}_{\pm 0.014}$ | $0.636_{\pm 0.002}$ | $0.655_{\pm 0.007}$ |
| | SA-QED | $0.558_{\pm 0.021}$ | $0.582_{\pm 0.004}$ | $0.594_{\pm 0.006}$ | $\mathbf{0.607}_{\pm 0.008}$ | $0.587_{\pm 0.011}$ |
| | ALL | $0.664_{\pm 0.020}$ | $0.643_{\pm 0.012}$ | $0.591_{\pm 0.035}$ | $0.658_{\pm 0.002}$ | $\mathbf{0.665}_{\pm 0.007}$ |

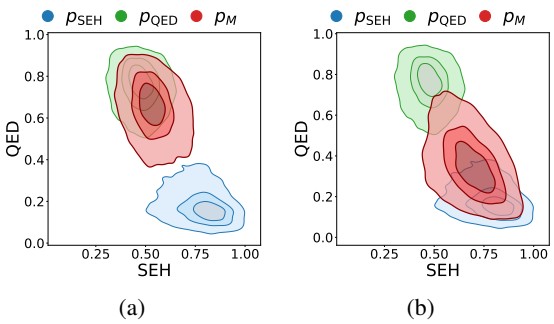

*Figure 4.* Density visualization of $p_{\text{SEH}}$, $p_{\text{QED}}$, and the composed distribution $p_M$ induced by our mixing policy via scalarization on fragment-based molecule generation. Each distribution is estimated from 5,000 samples. We vary the weights as (a) $p_M \propto (0.3 \cdot R_{\text{SEH}} + 0.7 \cdot R_{\text{QED}})^{32}$ and (b) $p_M \propto (0.7 \cdot R_{\text{SEH}} + 0.3 \cdot R_{\text{QED}})^{32}$.

resented as a connected graph, and actions consist of adding new nodes or edges to the graph and setting their attributes. We consider three reward functions: GAP, derived from the HOMO–LUMO gap predicted by an MXMNet (Zhang et al., 2020) proxy trained on the QM9 dataset (transformed so that smaller gaps yield higher rewards, clipped to $[0, 2]$); and the aforementioned SA and QED.

As in the synthetic experiments, we evaluate both scalarization and logical operators. For scalarization, we use 10 and 128 evenly spaced preference vectors for two- and three-objective settings, respectively. For logical operators, we test various combinations applied to the base rewards.

**Baselines.** For scalarization, we compare against two preference-conditioned GFlowNets, MOGFN (Jain et al., 2023b) and HN-GFN (Zhu et al., 2023). We additionally include OP-GFN (Chen & Mauch, 2024) for the hypervolume comparison. Since OP-GFN targets Pareto-optimal samples without conditioning on preference vectors, it is not directly comparable on per-preference reward metrics. For logical operators, we compare against the classifier guidance approach of Garipov et al. (2023). We report two variants of our method, Model $F$ and DB $F$, corresponding to the two routes for computing the reaching probabilities introduced in Section 4.1.

**Evaluation metrics.** Unlike in the grid experiments, neither the target nor the induced distributions can be computed

due to the large state space. For scalarization, we sample 128 candidates per preference vector, select the top-10 by scalarized reward, and compute their mean reward and diversity (1 - Tanimoto similarity). We additionally report the hypervolume of the generated molecules, a standard multi-objective optimization metric capturing both convergence and spread (Jain et al., 2023b). For logical operators, following Garipov et al. (2023), we categorize 5,000 samples into $2^3 = 8$ bins based on whether each reward is above or below a threshold and report the percentage in each bin.

For the temperature parameter, we set $\beta = 32$ for scalarization and logical operators. Further details are provided in Section B.2.

### 6.2. Results

**Scalarization.** Tables 3 and 4 present the scalarization results for both fragment-based and atom-based molecule generation. Table 3 reports the average reward and diversity over preference vectors, and Table 4 reports the hypervolume of the generated molecules. Across both domains, our method outperforms the preference-conditioned baselines

*Table 5.* Results of logical operators on molecule generation tasks (mean $\pm$ std over 3 seeds). We report the percentage of 5,000 samples falling into the *target bin*: for harmonic mean ($\otimes$), where all composed objectives are high; for contrast ($\CIRCLE$), where only the first objective is high.

| | Fragment-based | | | | Atom-based (QM9) | | |
|---|---|---|---|---|---|---|---|
| | Classifier guidance | Ours (Model $F$) | Ours (DB $F$) | | Classifier guidance | Ours (Model $F$) | Ours (DB $F$) |
| *Harmonic mean ($\otimes$)* | | | | *Harmonic mean ($\otimes$)* | | | |
| $p_{\text{SEH}} \otimes p_{\text{SA}}$ | $81_{\pm 2}$ | $87_{\pm 3}$ | $\mathbf{88}_{\pm 3}$ | $p_{\text{GAP}} \otimes p_{\text{SA}}$ | $78_{\pm 4}$ | $\mathbf{87}_{\pm 1}$ | $86_{\pm 2}$ |
| $p_{\text{SEH}} \otimes p_{\text{QED}}$ | $80_{\pm 4}$ | $87_{\pm 2}$ | $\mathbf{88}_{\pm 2}$ | $p_{\text{GAP}} \otimes p_{\text{QED}}$ | $76_{\pm 2}$ | $\mathbf{79}_{\pm 10}$ | $78_{\pm 2}$ |
| $p_{\text{SEH}} \otimes p_{\text{SA}} \otimes p_{\text{QED}}$ | $40_{\pm 3}$ | $65_{\pm 2}$ | $\mathbf{66}_{\pm 3}$ | $p_{\text{GAP}} \otimes p_{\text{SA}} \otimes p_{\text{QED}}$ | $51_{\pm 3}$ | $61_{\pm 8}$ | $\mathbf{63}_{\pm 2}$ |
| *Contrast ($\CIRCLE$)* | | | | *Contrast ($\CIRCLE$)* | | | |
| $p_{\text{SEH}} \CIRCLE p_{\text{SA}}$ | $68_{\pm 2}$ | $\mathbf{76}_{\pm 1}$ | $\mathbf{76}_{\pm 1}$ | $p_{\text{GAP}} \CIRCLE p_{\text{SA}}$ | $55_{\pm 3}$ | $55_{\pm 3}$ | $\mathbf{56}_{\pm 4}$ |
| $p_{\text{SEH}} \CIRCLE p_{\text{QED}}$ | $83_{\pm 5}$ | $\mathbf{85}_{\pm 5}$ | $85_{\pm 4}$ | $p_{\text{GAP}} \CIRCLE p_{\text{QED}}$ | $79_{\pm 2}$ | $\mathbf{85}_{\pm 2}$ | $84_{\pm 2}$ |
| $p_{\text{SEH}} \CIRCLE p_{\text{SA}} \CIRCLE p_{\text{QED}}$ | $57_{\pm 2}$ | $\mathbf{64}_{\pm 3}$ | $63_{\pm 3}$ | $p_{\text{GAP}} \CIRCLE p_{\text{SA}} \CIRCLE p_{\text{QED}}$ | $45_{\pm 3}$ | $49_{\pm 3}$ | $\mathbf{50}_{\pm 4}$ |

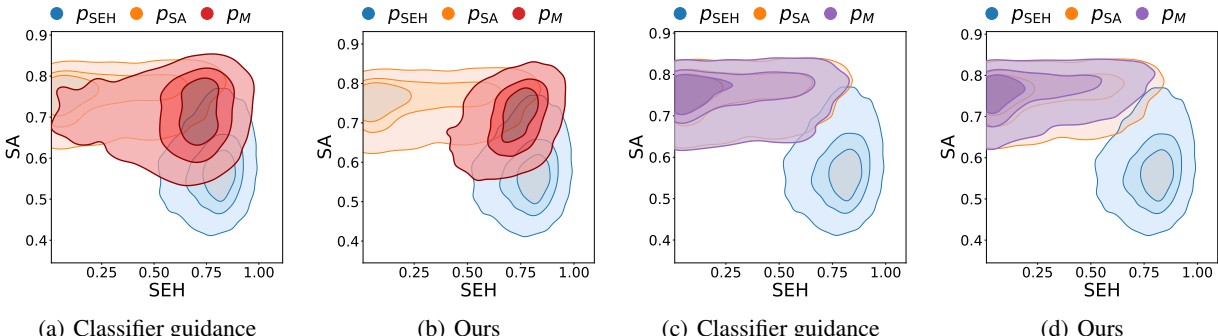

(a) Classifier guidance     (b) Ours     (c) Classifier guidance     (d) Ours

*Figure 5.* Distributions $p_M$ induced by classifier-guidance and our mixing policy for fragment-based molecule generation. Each distribution is estimated from 5,000 samples. Base distributions $p_{\text{SEH}}$ and $p_{\text{SA}}$ are shown alongside the composed distribution. (a) Classifier-guidance on $p_{\text{SEH}} \otimes p_{\text{SA}}$. (b) Ours on $p_{\text{SEH}} \otimes p_{\text{SA}}$. (c) Classifier-guidance on $p_{\text{SA}} \CIRCLE p_{\text{SEH}}$. (d) Ours on $p_{\text{SA}} \CIRCLE p_{\text{SEH}}$.

on reward in most objective combinations while maintaining comparable diversity. On hypervolume, our method also attains the best result in most objective combinations. Overall, our approach matches or exceeds the performance of baselines that require joint training for each objective combination. Fig. 4 shows scalarization with different preference vectors $\omega$. As $\omega$ shifts toward one objective, the induced distribution moves toward the corresponding high-reward region. Additional results, including IGD+, Pareto Count, and an ensemble ablation across SubTB- and TB-trained base GFlowNets, are provided in Tables A3 to A5.

**Logical operators.** Table 5 reports the percentage of samples in the target bin for each composition of the models. For the harmonic mean operator, the target bin is where all composed objectives are high; for the contrast operator, it is where only the first objective is high and the remaining objectives are low. Our method consistently achieves high target bin percentages, outperforming classifier guidance.

Tables A6 to A8 present the detailed distribution of samples across all reward bins for the base GFlowNets as well as their compositions with our mixing policy and classifier

guidance. In single-objective models, samples concentrate on high values of their respective training rewards, with variation across the other rewards. For the harmonic mean operator, our mixing policy successfully concentrates samples on molecules scoring high in all composed objectives, and extending to three objectives further amplifies this effect. For the contrast operator, our mixing policy effectively emphasizes the first objective while suppressing the others, isolating regions distinctive to the first component. These results confirm that our mixing policy correctly induces the intended compositional behaviors across both operators.

Fig. 5 visualizes the reward distributions induced by the classifier-guidance and our mixing policies. Figs. 5(a) and 5(b) show the results on the harmonic mean of SEH and SA ($p_{\text{SEH}} \otimes p_{\text{SA}}$). Our mixing policy produces samples concentrated in regions where both rewards are high, whereas the classifier-guidance often produces high-SA but low-SEH samples. Figs. 5(c) and 5(d) present the results on the contrast of SA and SEH ($p_{\text{SA}} \CIRCLE p_{\text{SEH}}$). Our mixing policy successfully isolates samples scoring high only on SA while suppressing the high-SEH region, as intended by the contrast operator. Conversely, the classifier guidance pro-

*Table 6.* Inference cost comparison on molecule generation tasks. We report the average generation time (ms) and trajectory length over 1,000 samples. We use preference vectors $\omega \in \{(0.5, 0.5), (0.33, 0.33, 0.34)\}$ for scalarization, and the harmonic mean for logical operators.

| | # Obj. | | Fragment-based | | Atom-based (QM9) | |
|---|---|---|---|---|---|---|
| | | | Time (ms) | Traj. Len. | Time (ms) | Traj. Len. |
| *Scalarization* | 2 | MOGFN | 21.62 | 26.0 | 38.41 | 31.5 |
| | | HN-GFN | 21.35 | 26.0 | 38.90 | 32.8 |
| | | OP-GFN | 21.78 | 26.0 | 50.51 | 41.7 |
| | | Ours (Model $F$) | 29.89 | 26.0 | 54.68 | 30.2 |
| | | Ours (DB $F$) | 32.79 | 26.0 | 57.88 | 30.1 |
| | 3 | MOGFN | 21.94 | 25.9 | 40.13 | 32.8 |
| | | HN-GFN | 21.49 | 26.0 | 29.93 | 24.7 |
| | | OP-GFN | 22.10 | 26.0 | 41.30 | 37.0 |
| | | Ours (Model $F$) | 33.57 | 26.0 | 62.95 | 31.5 |
| | | Ours (DB $F$) | 37.36 | 26.0 | 68.30 | 31.8 |
| *Logical* | 2 | Classifier guidance | 1900.71 | 25.9 | 1765.92 | 27.8 |
| | | Ours (Model $F$) | 29.62 | 26.0 | 55.88 | 31.4 |
| | | Ours (DB $F$) | 32.47 | 26.0 | 57.55 | 30.3 |
| | 3 | Classifier guidance | 2107.68 | 25.9 | 1853.85 | 31.0 |
| | | Ours (Model $F$) | 32.50 | 26.0 | 63.09 | 32.6 |
| | | Ours (DB $F$) | 36.34 | 26.0 | 66.08 | 31.7 |

*Table 7.* Validity of generated molecules on the atom-based molecule generation task. We report the percentage of valid molecules among 1,000 generated samples, averaged over 3 seeds (mean ± std). Note that the base models achieve validity ($p_{\text{GAP}}$: $98.5_{\pm 0.5}$, $p_{\text{SA}}$: $90.0_{\pm 14.5}$, $p_{\text{QED}}$: $97.2_{\pm 2.6}$).

| | Classifier guidance | Ours (Model $F$) | Ours (DB $F$) |
|---|---|---|---|
| $p_{\text{GAP}} \otimes p_{\text{SA}}$ | $93.8_{\pm 2.9}$ | $97.0_{\pm 2.5}$ | $\mathbf{97.8}_{\pm 1.4}$ |
| $p_{\text{GAP}} \otimes p_{\text{SA}} \otimes p_{\text{QED}}$ | $90.4_{\pm 6.5}$ | $95.6_{\pm 2.6}$ | $\mathbf{97.2}_{\pm 1.8}$ |
| $p_{\text{SA}} \oplus p_{\text{GAP}}$ | $92.3_{\pm 10.3}$ | $98.2_{\pm 0.7}$ | $\mathbf{98.3}_{\pm 0.8}$ |
| $p_{\text{SA}} \oplus p_{\text{GAP}} \oplus p_{\text{QED}}$ | $91.3_{\pm 12.2}$ | $\mathbf{92.3}_{\pm 11.0}$ | $90.9_{\pm 13.9}$ |

duces a distribution that nearly coincides with the base SA distribution ($p_{\text{SA}}$), failing to achieve the subtraction effect.

### 6.3. Analysis

**Inference costs.** We evaluate inference costs in real-world experiments. We generate 1,000 samples and report the average generation time and trajectory length in Table 6. For scalarization, our method takes approximately 8–40 milliseconds longer per sample than MOGFN, HN-GFN, and OP-GFN, due to the overhead of querying multiple base models. The gap is larger on QM9 partly because our method generates longer trajectories. Despite this overhead, our method offers the flexibility to incorporate new objectives by training only the corresponding base GFlowNet while reusing existing ones. For logical operators, our method is approximately 30–65 times faster than classifier-guidance, while both methods generate trajectories of similar length. This is because classifier guidance must enumerate and evaluate all successor states at every step to compute the guidance term, whereas our method only requires computation at the current state. Notably, our implementation does not parallelize forward passes of base models. These results demonstrate that our approach incurs minimal inference overhead.

**Validity of generated molecules.** We evaluate the validity of generated molecules on QM9. We generate 1,000 samples and report the percentage of valid molecules for four representative logical operations in Table 7. Whereas classifier guidance trains an auxiliary classifier jointly over all objectives and steers sampling with it as an external guidance signal, our mixing policy directly composes base GFlowNets that are each trained in isolation on a single

objective. Despite this, our method maintains molecular validity, staying close to the base GFlowNets and performing on par with or better than classifier guidance. Overall, our mixing rule composes independently trained GFlowNets while preserving the molecular validity of the generated samples.

## 7. Conclusion

In this work, we propose a method for composing pre-trained GFlowNets via a simple mixing rule over learned forward policies. For linear scalarization, we theoretically prove that our mixing rule exactly recovers the target composition distribution. For more general nonlinear operators, we empirically verify that our method accurately approximates the target distribution in high-density regions most relevant to generation. Experiments on 2D grid and molecule generation tasks demonstrate that our method achieves competitive performance with existing baselines, requiring no additional training at the composition stage and incurring only minimal inference overhead. Our framework enables pre-trained GFlowNets to serve as reusable building blocks, composed to target diverse objective combinations at inference time.

## Impact Statement

This paper presents work whose goal is to advance the field of Machine Learning. The primary application of our method is molecular generation for drug discovery, where composing pre-trained GFlowNets can accelerate the exploration of candidates balancing multiple properties such as binding affinity, synthetic accessibility, and drug-likeness. As with any molecular generation technology, dual-use concerns exist; however, our method composes existing models rather than introducing fundamentally new capabilities, and the generation of candidate molecules remains only one step in a complex pipeline requiring synthesis and experimental validation. We encourage continued development of safety guidelines and best practices for responsible deployment of generative models in scientific discovery.

## Acknowledgements

This work was supported by the National Research Foundation of Korea (NRF) grant funded by the Korea government (MSIT) (RS-2024-00337955); by the Institute of Information & Communications Technology Planning & Evaluation (IITP) grants funded by the Korea government (MSIT) (RS-2024-00457882, Artificial Intelligence Research Hub Project; RS-2019-II191906, Artificial Intelligence Graduate School Program (POSTECH)); and by IITP under the Leading Generative AI Human Resources Development (IITP-2026-RS-2026-25546560) grant funded by the Korea government (MSIT).

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

# A. Theoretical Details

## A.1. Exact Realization for Scalarization

We prove that for scalarization with $\beta = 1$, our mixing rule exactly realizes the target distribution.

**Setup.** Let $(\mathcal{S}, \mathcal{A})$ be the directed acyclic state graph of a GFlowNet with initial state $s_0$, sink state $s_f$, and terminating states $\mathcal{X} \subset \mathcal{S}$. For each objective $i \in [k]$, we have a pre-trained GFlowNet with forward policy $p_{i,F}(\cdot|\cdot)$, partition function $Z_i > 0$, and terminating state distribution

$$p_i(x) = \frac{R_i(x)}{Z_i}, \qquad R_i(x) \geq 0, \ x \in \mathcal{X}. \tag{10}$$

We write $u_i(s)$ for the reaching probability of $s$ under $p_{i,F}$, satisfying the recursion Eq. (2). Let the scalarization weights be $\omega_i \geq 0$ and define

$$v_i = \frac{\omega_i Z_i}{\sum_{j=1}^k \omega_j Z_j}, \qquad \sum_{i=1}^k v_i = 1. \tag{11}$$

**Target terminating state distribution.** The mixture terminating state distribution satisfies

$$p_M(x) := \sum_{i=1}^k v_i p_i(x) = \frac{\sum_{i=1}^k \omega_i R_i(x)}{\sum_{j=1}^k \omega_j Z_j} \propto \sum_{i=1}^k \omega_i R_i(x). \tag{12}$$

**Composed forward policy.** Define the mixed reaching probability

$$u_M(s) := \sum_{i=1}^k v_i u_i(s), \tag{13}$$

and the composed forward policy

$$p_{M,F}(s'|s) := \sum_{i=1}^k \frac{v_i u_i(s)}{u_M(s)} p_{i,F}(s'|s) \quad \text{for all } s \in \mathcal{S} \setminus \{s_f\}. \tag{14}$$

Note that Eq. (14) is a special case of the general mixing policy Eq. (4) for scalarization with $\beta = 1$: since $\mathcal{G}$ reduces to a weighted sum, $N_M(s) = \left( \sum_j \omega_j Z_j \right) u_M(s)$, and this constant factor cancels in the policy.

We now prove that $p_{M,F}$ is a valid policy and induces the desired mixture distribution.

**Lemma A.1** (Normalization). *For any $s \neq s_f$, $\sum_{s':(s \to s') \in \mathcal{A}} p_{M,F}(s'|s) = 1$.*

*Proof.* Using Eq. (14) and the fact that each $p_{i,F}(\cdot|s)$ is a probability distribution,

$$\sum_{s'} p_{M,F}(s'|s) = \sum_{i=1}^k \frac{v_i u_i(s)}{u_M(s)} \underbrace{\sum_{s'} p_{i,F}(s'|s)}_{=1} = \frac{\sum_{i=1}^k v_i u_i(s)}{u_M(s)} = 1,$$

where the last equality uses Eq. (13). $\qquad \square$

**Lemma A.2** (Mixed reaching probabilities). *Let $u_M$ be defined by Eq. (13). Then $u_M$ coincides with the reaching probability induced by $p_{M,F}$:*

$$u_M(s_0) = 1,$$
$$u_M(s) = \sum_{s_*:(s_* \to s) \in \mathcal{A}} u_M(s_*) \, p_{M,F}(s|s_*) \quad \forall s \in \mathcal{S} \setminus \{s_0\}.$$

*Proof.* The base case $u_M(s_0) = \sum_i v_i u_i(s_0) = \sum_i v_i = 1$ follows from $u_i(s_0) = 1$ and Eq. (11). For $s \neq s_0$, substitute Eq. (14):

$$\sum_{s_*} u_M(s_*) \, p_{M,F}(s|s_*)$$

$$= \sum_{s_*} \Big( \sum_{j=1}^{k} v_j u_j(s_*) \Big) \Big( \sum_{i=1}^{k} \frac{v_i u_i(s_*)}{u_M(s_*)} \, p_{i,F}(s|s_*) \Big)$$

$$= \sum_{i=1}^{k} v_i \sum_{s_*} u_i(s_*) \, p_{i,F}(s|s_*) = \sum_{i=1}^{k} v_i u_i(s) = u_M(s),$$

using the reaching probability recursion Eq. (2) in the penultimate equality. □

**Lemma A.3** (State distribution factorization). *For each $i$, the state probability satisfies*

$$p_i(s) = u_i(s)\,\rho_i(s) \quad and$$
$$p_M(s) = u_M(s)\,\rho_M(s), \tag{15}$$

*where $\rho_i(s)$ (resp. $\rho_M(s)$) is the probability of terminating the trajectory when at state $s$ under $p_{i,F}$ (resp. $p_{M,F}$).*

*Proof.* This is the standard GFlowNet factorization: the probability of being at $s$ equals the probability of reaching $s$ times the probability of terminating from $s$. The equality for $p_M$ follows since $p_{M,F}$ is a valid forward policy by Lemma A.1. □

**Theorem A.4** (Correctness of the composed forward policy). *Let $v_i$ be given by Eq. (11) and $p_{M,F}$ by Eq. (14). Then, for all states $s$,*

$$p_M(s) = \sum_{i=1}^{k} v_i p_i(s). \tag{16}$$

*Consequently, the induced terminating state distribution satisfies $p_M(x) = \sum_i v_i p_i(x) \propto \sum_i \omega_i R_i(x)$ for $x \in \mathcal{X}$.*

*Proof.* By Lemma A.3 and the definition of $p_{M,F}$,

$$p_M(s) = u_M(s)\,\rho_M(s)$$

$$= u_M(s) \sum_{i=1}^{k} \frac{v_i u_i(s)}{u_M(s)} \, \rho_i(s)$$

$$= \sum_{i=1}^{k} v_i u_i(s)\,\rho_i(s)$$

$$= \sum_{i=1}^{k} v_i p_i(s),$$

where the second equality follows from applying Eq. (14) to the termination probability, and the last step uses Eq. (15) for each $i$. Evaluating Eq. (16) on terminating states $x \in \mathcal{X}$ yields $p_M(x) = \sum_i v_i p_i(x)$; substituting the definitions proves the proportionality claim via Eq. (12). □

**Remarks.**    (i) The result holds for any non-negative $\omega_i$; they need not sum to 1. (ii) If estimated partition functions $\hat{Z}_i$ are used in place of $Z_i$, the sampler targets the scalarized reward up to a global rescaling, which does not affect the induced distribution. (iii) The support of $p_M$ equals the union of supports of the $p_i$; no extra coverage assumptions are required beyond a shared state graph. (iv) This exactness result holds specifically for scalarization with $\beta = 1$. For general nonlinear composition operators or $\beta \neq 1$, the distortion factor $\delta(x) = u_M(x)/N_M(x)$ introduced in Section 4.2 characterizes the gap between the realized and target distributions.

### A.2. Derivation of Reaching Probability

We derive an expression for the reaching probability and describe two equivalent routes for obtaining the state flow $F_i(s)$ that make our framework compatible with every standard GFlowNet training objective.

**Training objectives and model parameterization.** GFlowNet training objectives differ in what quantities they parameterize. Flow Matching (FM) (Bengio et al., 2021) parameterizes the state flow $F(s)$ for all states $s \in \mathcal{S}$, from which the forward policy $p_F$ is derived. Detailed Balance (DB) (Bengio et al., 2023) parameterizes the forward policy $p_F$ and backward policy $p_B$ alongside $F(s)$. In both cases, the partition function is $Z = F(s_0)$. Sub-Trajectory Balance (SubTB) (Madan et al., 2023) similarly parameterizes the state flow $F(s)$, $p_F$, and $p_B$. In contrast, Trajectory Balance (TB) (Malkin et al., 2022) only parameterizes $p_F$, $p_B$, and $Z$, without explicitly learning the state flow $F(s)$. Our mixing rule requires the reaching probability $u_i(s)$, which can be computed from $F_i(s)$ and $Z_i$ (as we derive below). The problem thus reduces to obtaining $F_i(s)$. We present two routes: a **Model** $F$ route that directly reads $F_i(s)$ from a learned state flow (available for FM, DB, and SubTB), and a **DB** $F$ route that reconstructs $F_i(s)$ recursively from $p_F$ and $p_B$ via the detailed balance identity, making our framework compatible with any standard objective including TB.

#### A.2.1. DERIVATION OF $u_i(s) = F_i(s)/Z_i$

For any state $s \in \mathcal{S}$, the state flow $F_i(s)$ represents the total flow passing through $s$:

$$F_i(s) = \sum_{\tau : s \in \tau} F_i(\tau), \tag{17}$$

where the sum is over all complete trajectories $\tau$ that pass through $s$, and $F_i(\tau)$ is the flow along trajectory $\tau$.

Since the total flow equals the partition function, $\sum_\tau F_i(\tau) = Z_i$, we can write:

$$\frac{F_i(s)}{Z_i} = \frac{\sum_{\tau : s \in \tau} F_i(\tau)}{\sum_\tau F_i(\tau)} = \sum_{\tau : s \in \tau} \frac{F_i(\tau)}{Z_i} = \sum_{\tau : s \in \tau} p_i(\tau), \tag{18}$$

where $p_i(\tau) = F_i(\tau)/Z_i$ is the probability of trajectory $\tau$.

By definition Eq. (1), $u_i(s) = \sum_{\tau \in \mathcal{T}_{s_0,s}} p_i(\tau)$, where $\mathcal{T}_{s_0,s}$ denotes the set of trajectories from $s_0$ to $s$. Since the DAG structure ensures that every trajectory starting from $s$ eventually reaches $s_f$ with probability one, this equals $\sum_{\tau : s \in \tau} p_i(\tau)$. Therefore,

$$u_i(s) = \frac{F_i(s)}{Z_i} \quad \text{for all } s \in \mathcal{S}. \tag{19}$$

Given this identity, computing $u_i(s)$ reduces to obtaining $F_i(s)$, which we recover through the two routes below.

#### A.2.2. MODEL $F$: DIRECT READ FROM THE LEARNED STATE FLOW

When the base GFlowNet is trained with FM, DB, or SubTB, the state flow $F_i(s)$ is an explicit output of the learned model for every state $s \in \mathcal{S}$, and $Z_i = F_i(s_0)$. Plugging these learned quantities directly into Eq. (19) yields $u_i(s)$, requiring no further computation beyond a forward pass of the flow network.

#### A.2.3. DB $F$: RECURSIVE COMPUTATION VIA THE DETAILED BALANCE IDENTITY

The detailed balance identity holds at convergence for any well-trained GFlowNet, regardless of the training objective, allowing us to recover $u_i(s)$ from the forward and backward policies alone. For every edge $s \to s'$:

$$F_i(s)\, p_{i,F}(s'|s) = F_i(s')\, p_{i,B}(s|s'), \tag{20}$$

which can be rearranged as

$$F_i(s') = F_i(s) \cdot \frac{p_{i,F}(s'|s)}{p_{i,B}(s|s')}. \tag{21}$$

Unrolling this recursion along any trajectory $s_0 \to s_1 \to \cdots \to s_t$ starting from $F_i(s_0) = Z_i$ gives

$$F_i(s_t) = Z_i \prod_{j=1}^{t} \frac{p_{i,F}(s_j|s_{j-1})}{p_{i,B}(s_{j-1}|s_j)}. \tag{22}$$

Substituting Eq. (22) into Eq. (19) yields a closed-form expression for the reaching probability that depends only on $p_{i,F}$ and $p_{i,B}$:

$$u_i(s_t) = \frac{F_i(s_t)}{Z_i} = \prod_{j=1}^{t} \frac{p_{i,F}(s_j|s_{j-1})}{p_{i,B}(s_{j-1}|s_j)}. \tag{23}$$

Because Eq. (23) references neither $F_i(\cdot)$ nor $Z_i$ on the right-hand side, it applies to any base GFlowNet trained with a standard objective, including TB. Moreover, the product form admits on-the-fly computation during trajectory rollout: as the sampler traverses $s_0, s_1, \ldots, s_t$, the ratio $p_{i,F}(s_j|s_{j-1})/p_{i,B}(s_{j-1}|s_j)$ is evaluated at each step and accumulated multiplicatively, yielding $u_i(s_t)$ with no additional model queries beyond those already performed for sampling.

### A.3. Derivation of $L_1$ Decomposition

We derive the decomposition of the $L_1$ distance between the induced distribution $p_M(x)$ and the target distribution $p_M^*(x)$ (Eq. (9) in the main text).

**Setup.** Given $k$ pre-trained GFlowNets with terminating distributions $p_i(x) \propto R_i(x)$, recall from Section 4.2 that the distribution induced by our mixing policy takes the form:

$$p_M(x) = \frac{u_M(x)}{N_M(x)} \cdot \mathcal{G}(p_1(x), \ldots, p_k(x)), \tag{24}$$

where $\delta(x) := u_M(x)/N_M(x)$ is the distortion factor.

The target distribution is defined as:

$$p_M^*(x) = \frac{\mathcal{G}(p_1(x), \ldots, p_k(x))}{Z_M}, \tag{25}$$

where $Z_M = \sum_x \mathcal{G}(p_1(x), \ldots, p_k(x))$ is the partition function.

**Derivation.** The $L_1$ distance between the induced and target distributions is:

$$L_1 = \sum_{x \in \mathcal{X}} |p_M(x) - p_M^*(x)|. \tag{26}$$

Substituting the expressions for $p_M(x)$ and $p_M^*(x)$:

$$\begin{aligned} L_1 &= \sum_{x \in \mathcal{X}} |p_M(x) - p_M^*(x)| \\ &= \sum_{x \in \mathcal{X}} \left| \frac{u_M(x)}{N_M(x)} \cdot \mathcal{G}(p_1(x), \ldots, p_k(x)) - \frac{1}{Z_M} \cdot \mathcal{G}(p_1(x), \ldots, p_k(x)) \right|. \end{aligned} \tag{27}$$

Using the definition of the distortion factor $\delta(x) = u_M(x)/N_M(x)$:

$$L_1 = \sum_{x \in \mathcal{X}} \left| \delta(x) \cdot \mathcal{G}(p_1(x), \ldots, p_k(x)) - \frac{1}{Z_M} \cdot \mathcal{G}(p_1(x), \ldots, p_k(x)) \right|. \tag{28}$$

Since $\mathcal{G}(p_1(x), \ldots, p_k(x)) \geq 0$ for all $x$ (as it is defined through non-negative reward functions and probability distributions), we can factor it out of the absolute value:

$$L_1 = \sum_{x \in \mathcal{X}} \left| \delta(x) - \frac{1}{Z_M} \right| \cdot \mathcal{G}(p_1(x), \ldots, p_k(x)). \tag{29}$$

# B. Additional Experimental Details

## B.1. Synthetic Experiments

**Reward functions.**   We use five reward functions (Fig. A1(a)): Shubert, Currin, Sphere, and Branin from standard benchmark functions for global optimization (Jamil & Yang, 2013), and a Diagonal sigmoid reward. All rewards are normalized to $[0, 1]$. We additionally use three synthetic Circle rewards (Fig. A1(b)). Each Circle reward defines a Gaussian (mixture) density inside a circular region of radius $0.6$ and assigns a constant background reward $0.1$ outside the circle.

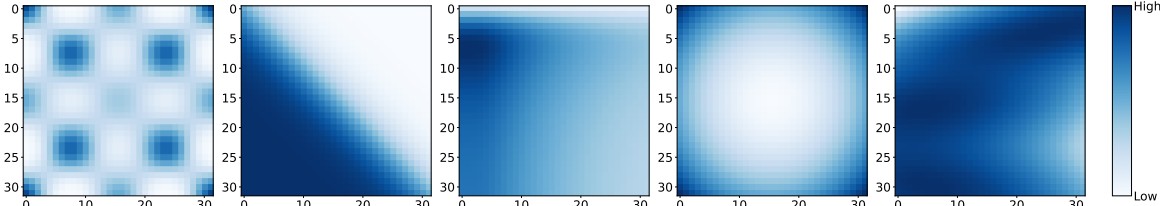

(a) Five rewards on the $32 \times 32$ grid. From left to right: Shubert, Diagonal, Currin, Sphere, and Branin.

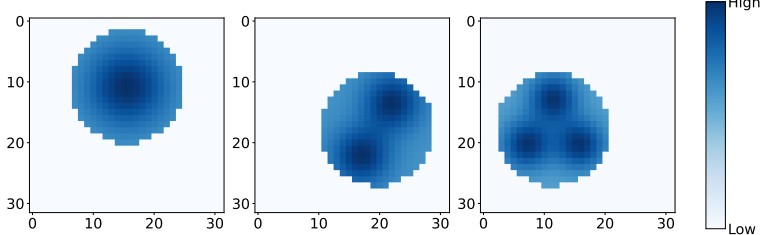

(b) Circle rewards on the $32 \times 32$ grid. From left to right: Circle1–3.

*Figure A1.* Visualization of reward functions on the $32 \times 32$ grid domain used in our experiments.

**Scalarization (Mix-$k$).**   We define scalarization targets using weighted-sum rewards over a subset of $k$ objectives. Table A1 lists the objective subsets used in our experiments.

*Table A1.* Objective subsets used for Mix-$k$ scalarization (Fig. A1(a)).

|       | Objectives |
|-------|------------|
| Mix2  | Shubert, Diagonal |
| Mix3  | Shubert, Diagonal, Currin |
| Mix4  | Shubert, Diagonal, Currin, Sphere |
| Mix5  | Shubert, Diagonal, Currin, Sphere, Branin |

**Model details and hyperparameters.**   All GFlowNets are trained using the Sub-Trajectory Balance (SubTB) (Madan et al., 2023) objective with geometric within-trajectory weighting ($\lambda = 2.0$). Each state is encoded as a 64-dimensional vector formed by concatenating two 32-dimensional one-hot coordinate vectors (one for each axis). The forward policy $p_F$ and backward policy $p_B$ are MLPs with 2 hidden layers of 64 units and ReLU (Nair & Hinton, 2010) activations, outputting logits over 3 actions (right, down, terminate) and 2 actions (left, up), respectively. The flow estimator $F$ uses a single hidden layer of 64 units and outputs $\log F(s)$. We train for 20,000 iterations using Adam (Kingma & Ba, 2015) with learning rate $10^{-3}$, batch size 128, and $\epsilon$-greedy exploration rate 0.05. We additionally use a replay buffer of size 10,000.

For the multi-objective baselines (MOGFN and HN-GFN), preferences are sampled from $\mathrm{Dirichlet}(\alpha = 1.0)$ during training. For MOGFN, the state and preference vector are encoded separately by two MLPs: the state encoder has 2 hidden layers of 64 units (output 64-dim), and the preference encoder has 2 hidden layers of 16 units (output 16-dim). The two embeddings are concatenated and passed through a 2-hidden-layer MLP output head. For HN-GFN (Zhu et al., 2023), the state is first encoded by an MLP with 2 hidden layers of 64 units. A HyperNetwork with a 3-layer ray MLP (hidden dimension 32) maps

the preference vector to the weights of a single linear output layer that maps the 64-dim state embedding to action logits. Both also use a replay buffer of size 10,000.

For the classifier-guided baseline (Garipov et al., 2023), the classifier is an MLP with 2 hidden layers of 64 units and LeakyReLU (Maas et al., 2013) activations. Training uses Adam with learning rate $10^{-3}$, batch size 64, and 15,000 iterations. The mixing parameter is sampled as $z \sim \mathcal{U}[-3.5, 3.5]$ with $\alpha = \sigma(z)$. We use an EMA target network (smoothing factor 0.995) and linearly increase the non-terminal loss weight $\gamma$ from 0 to 1 over the first 3,000 steps.

**Additional results.** Table A2 and Figs. A2 to A7 show additional results across diverse reward combinations.

*Table A2.* Extended results of Table 2. $L_1$ error on logical operators in a 2D grid. We evaluate harmonic mean ($\otimes$) and contrast ($\newmoon$) operators across different reward pairs.

| | Classifier guidance | Ensemble | Ours |
|---|---|---|---|
| *Harmonic mean ($\otimes$)* | | | |
| $p_{\text{Shubert}} \otimes p_{\text{Sphere}}$ | 0.158 | 0.145 | **0.136** |
| $p_{\text{Branin}} \otimes p_{\text{Sphere}}$ | **0.053** | 0.121 | 0.110 |
| $p_{\text{Shubert}} \otimes p_{\text{Diagonal}}$ | **0.142** | 0.190 | 0.180 |
| $p_{\text{Circle1}} \otimes p_{\text{Circle2}}$ | 0.397 | 0.453 | **0.229** |
| $p_{\text{Circle1}} \otimes p_{\text{Circle3}}$ | 0.266 | 0.476 | **0.189** |
| $p_{\text{Circle2}} \otimes p_{\text{Circle3}}$ | **0.108** | 0.472 | 0.301 |
| *Contrast ($\newmoon$)* | | | |
| $p_{\text{Shubert}} \newmoon p_{\text{Sphere}}$ | 0.175 | 0.175 | **0.111** |
| $p_{\text{Sphere}} \newmoon p_{\text{Shubert}}$ | 0.190 | 0.162 | **0.116** |
| $p_{\text{Branin}} \newmoon p_{\text{Sphere}}$ | 0.082 | 0.116 | **0.080** |
| $p_{\text{Sphere}} \newmoon p_{\text{Branin}}$ | **0.073** | 0.100 | 0.102 |
| $p_{\text{Diagonal}} \newmoon p_{\text{Shubert}}$ | **0.153** | 0.190 | 0.158 |
| $p_{\text{Shubert}} \newmoon p_{\text{Diagonal}}$ | 0.150 | 0.190 | **0.106** |
| $p_{\text{Circle1}} \newmoon p_{\text{Circle2}}$ | 0.245 | 0.356 | **0.231** |
| $p_{\text{Circle2}} \newmoon p_{\text{Circle1}}$ | 0.241 | 0.390 | **0.070** |
| $p_{\text{Circle1}} \newmoon p_{\text{Circle3}}$ | **0.122** | 0.413 | 0.163 |
| $p_{\text{Circle3}} \newmoon p_{\text{Circle1}}$ | 0.138 | 0.407 | **0.058** |
| $p_{\text{Circle2}} \newmoon p_{\text{Circle3}}$ | **0.098** | 0.327 | 0.135 |
| $p_{\text{Circle3}} \newmoon p_{\text{Circle2}}$ | **0.093** | 0.286 | 0.285 |

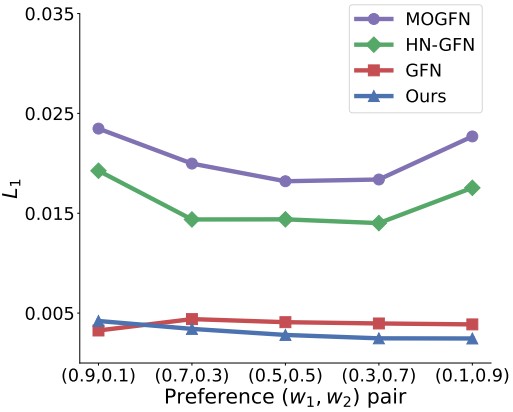

*Figure A2.* $L_1$ error for the scalarization of $p_{\text{Shubert}}$ and $p_{\text{Diagonal}}$ ($\beta = 1$) over five preference vectors $\boldsymbol{\omega} = (\omega_1, \omega_2)$. *GFN* denotes single-objective GFlowNets trained separately for each $\boldsymbol{\omega}$.

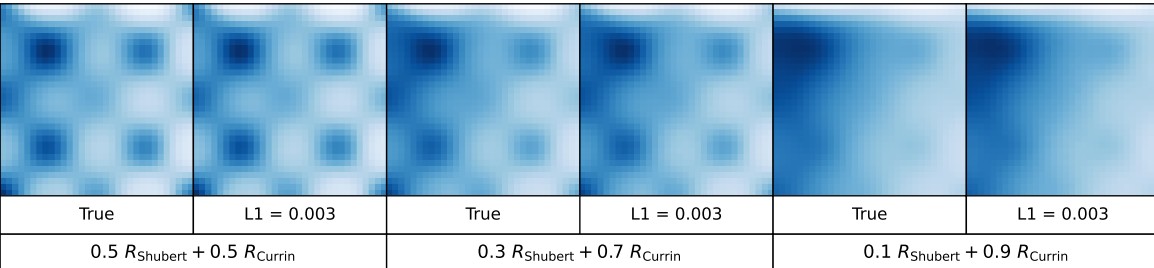

*Figure A3.* Density visualization of scalarization under different weight settings. For each setting, we show the ground-truth target distribution and the distribution induced by our method, with corresponding $L_1$ errors.

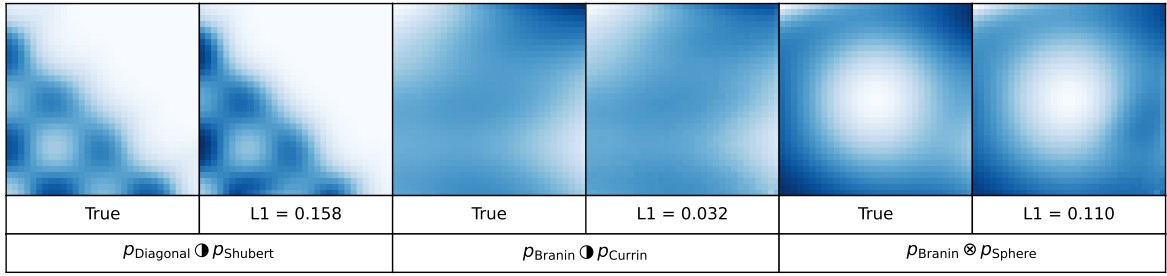

*Figure A4.* Density visualization of harmonic mean ($\otimes$) and contrast ($\circ$) compositions on pairs of benchmark reward functions. For each composition, we show the ground-truth target distribution and the distribution induced by our method, with corresponding $L_1$ errors.

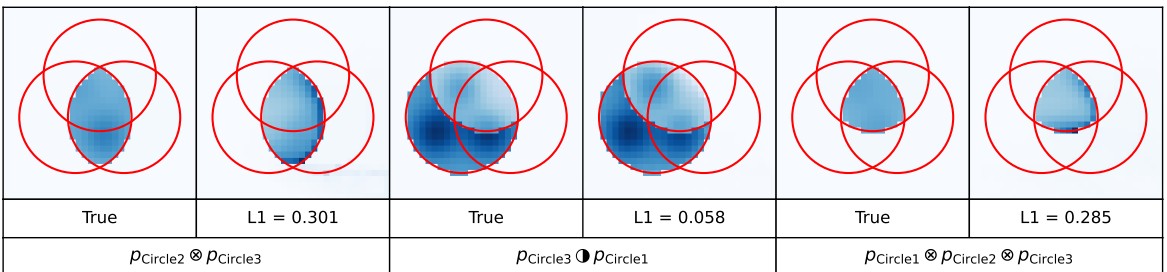

*Figure A5.* Density visualization of harmonic mean ($\otimes$) and contrast ($\circ$) compositions on Circle rewards. For each composition, we show the ground-truth target distribution and the distribution induced by our method, with corresponding $L_1$ errors.

## B.2. Real-world Experiments

**Reward functions.** For fragment-based generation, we use three reward functions. **SEH** is the predicted binding affinity to soluble epoxide hydrolase, computed using a pre-trained proxy model (Bengio et al., 2021), with raw values divided by 8 to normalize to approximately $[0, 1]$. **SA** is the synthetic accessibility score (Ertl & Schuffenhauer, 2009), computed using RDKit (Landrum, 2010) and transformed as $(10 - \mathrm{SA_{raw}})/9$ to map to $[0, 1]$ with higher values indicating easier synthesis. **QED** is the quantitative estimate of drug-likeness (Bickerton et al., 2012), computed using RDKit (Landrum, 2010), already in $[0, 1]$. For atom-based generation (QM9), we use **GAP**, the HOMO–LUMO gap predicted by an MXMNet (Zhang et al., 2020) proxy trained on QM9, rescaled and clipped to $[0, 2]$ so that smaller gaps yield higher rewards; SA and QED are computed as above.

**Model details and hyperparameters.** All GFlowNets use a Graph Transformer (Yun et al., 2019) backbone, with 6 layers and embedding dimension 128 for fragment-based generation (taking a junction tree representation as input where nodes correspond to fragments and edges encode attachment points), and 4 layers and embedding dimension 64 for atom-based generation (taking the current molecular graph as input with node features encoding atom types and edge features encoding bond types). Training uses the Sub-Trajectory Balance (SubTB) (Madan et al., 2023) objective with Adam (Kingma & Ba, 2015) (learning rate $5 \times 10^{-4}$, weight decay $10^{-8}$), batch size 64, random action probability 0.05, and 15,000 iterations.

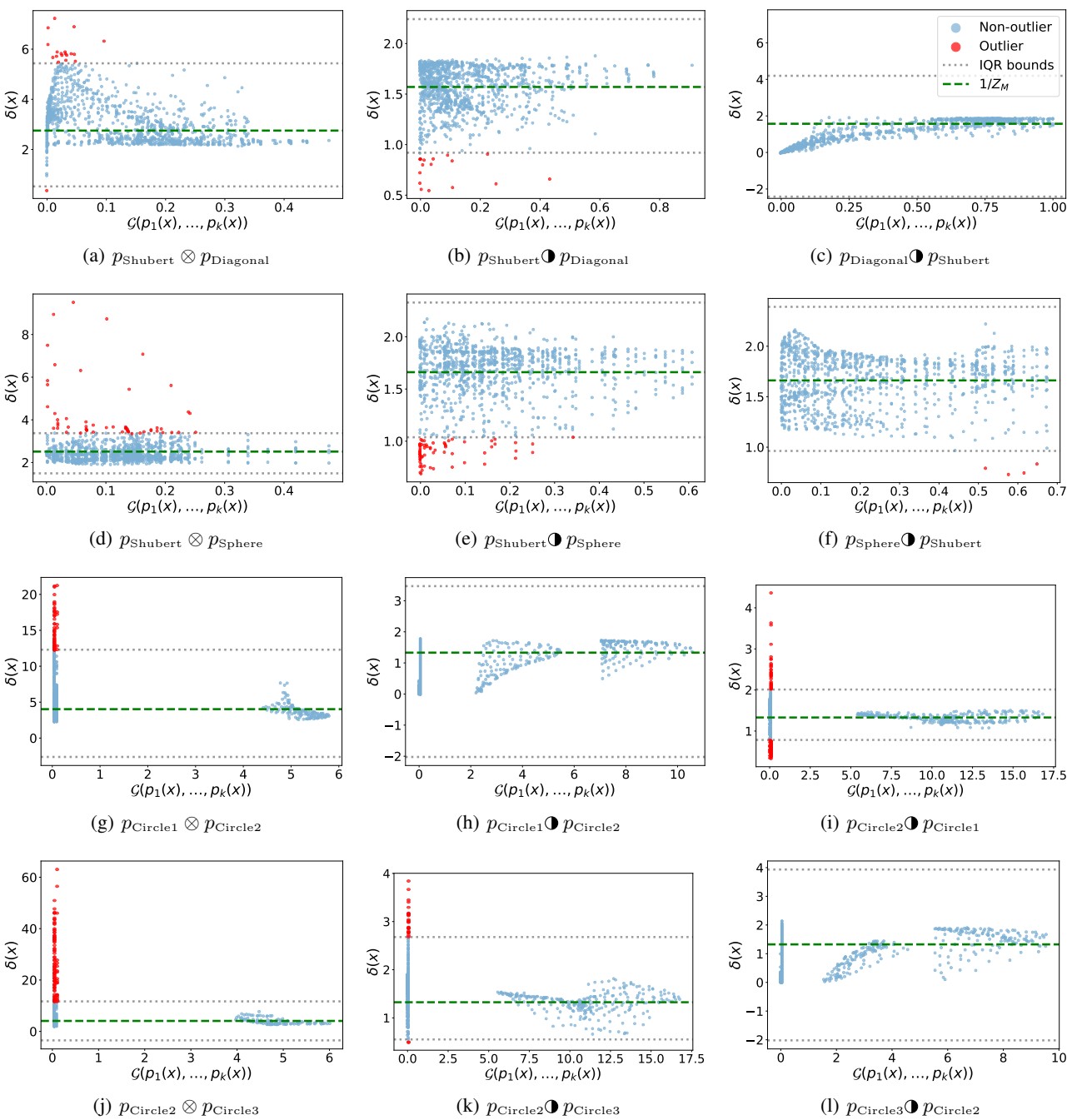

*Figure A6.* Extended results of Fig. 3. Distortion factor $\delta(x) = u_M(x)/N_M(x)$ vs. unnormalized target density $\mathcal{G}(p_1(x), \ldots, p_k(x))$ on the 2D grid domain. Red points indicate outliers, defined as states where $\delta(x)$ falls outside $[Q_1 - 1.5 \cdot \text{IQR}, Q_3 + 1.5 \cdot \text{IQR}]$ (gray dotted lines), where $Q_1$, $Q_3$ are the 25th/75th percentiles and $\text{IQR} = Q_3 - Q_1$. The green dashed line marks $1/Z_M$ with $Z_M = \sum_x \mathcal{G}(p_1(x), \ldots, p_k(x))$.

For multi-objective baselines (MOGFN and HN-GFN), preferences $\boldsymbol{\omega} \in \Delta^{k-1}$ are sampled from $\text{Dirichlet}(\alpha = 1.0)$ during training. MOGFN conditions the model by concatenating preferences to the graph-level embedding before the output heads. HN-GFN (Zhu et al., 2023) uses a HyperNetwork with a 3-layer MLP (hidden dimension 100) to generate policy network weights from preferences. OP-GFN (Chen & Mauch, 2024) replaces the standard trajectory balance loss with a cross-entropy over Pareto-dominance labels computed from each batch. Other hyperparameters match MOGFN.

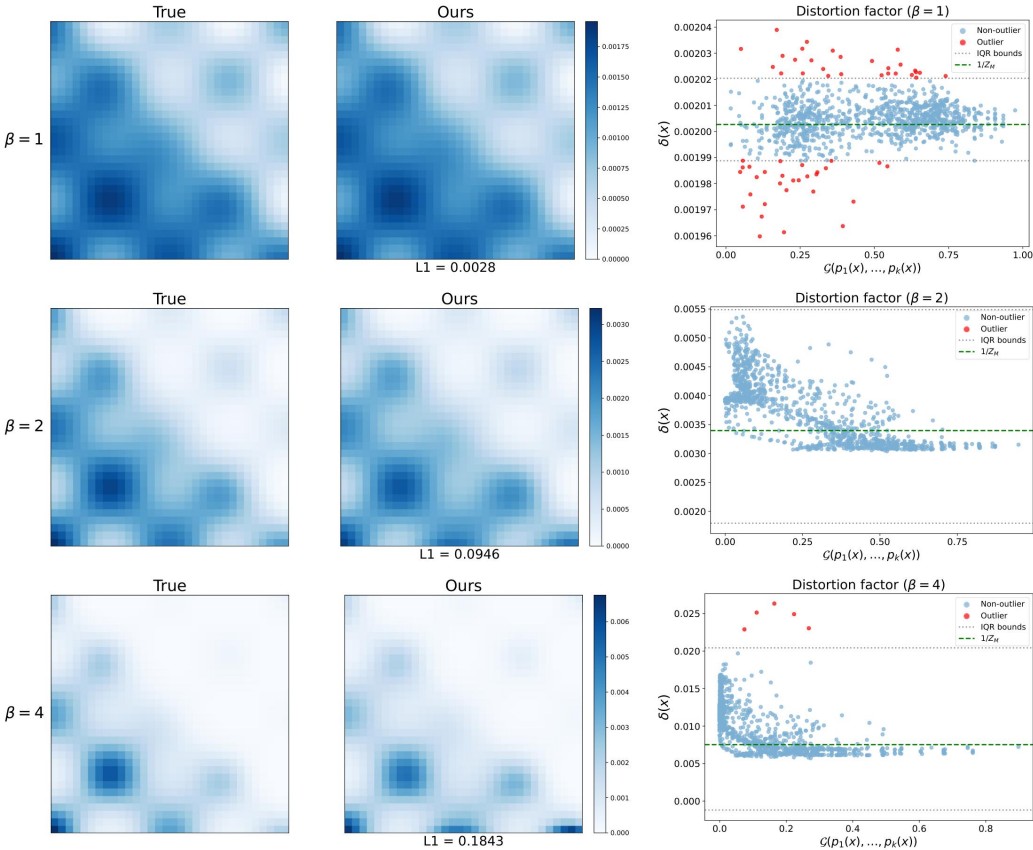

*Figure A7.* Qualitative results of scalarization with $\beta = 1, 2, 4$ on a 2D grid domain, where the reward is defined as $(0.5\ R_{\text{Shubert}} + 0.5\ R_{\text{Diagonal}})^{\beta}$. Pre-trained base models trained with reward exponent $\beta$ are composed using the reward-sharpened mixing policy Eq. (31). We visualize the density over the grid of the true distribution (left), Ours (middle), and the distortion factor analysis (right).

For the classifier-guided baseline (Garipov et al., 2023), we train 2-way and 3-way joint classifiers for two- and three-objective compositions, respectively. Each classifier is a Graph Transformer with 4 layers and 2 attention heads, with embedding dimension 128 for fragment-based and 64 for atom-based generation. Training uses Adam with learning rate $10^{-3}$, weight decay $10^{-6}$, batch size 8 per distribution, and 15,000 iterations. We use an EMA target network (smoothing factor 0.995) and linearly increase the non-terminal loss weight $\gamma$ from 0 to 1 over the first 4,000 steps. Training trajectories are sampled from pre-trained single-objective GFlowNets.

**Mixing reward-sharpened GFlowNets for scalarization.** For the scalarization target $p_M^*(x) \propto (\sum_i \omega_i R_i(x))^{\beta}$, we leverage pre-trained ingredient GFlowNets trained on $R_i^{\beta}$. In our mixing framework (Section 4.1), sampling from $p_M^*(x)$ requires ingredient models with terminating distributions $p_i(x) \propto R_i(x)$, which we can then mix as $p_{M,F} \propto (\sum_i \omega_i Z_i u_i p_{i,F})^{\beta}$. However, our ingredient GFlowNets are trained with reward exponent $\beta$, inducing terminating distributions $\tilde{p}_i(x) \propto R_i(x)^{\beta}$ with partition functions $\tilde{Z}_i = \sum_x R_i(x)^{\beta}$.

The unnormalized reward can be recovered as $R_i(x) = (\tilde{Z}_i \tilde{p}_i(x))^{1/\beta}$, since $\tilde{Z}_i \tilde{p}_i(x) = R_i(x)^{\beta}$. Substituting into the scalarization target:

$$p_M^*(x) \propto \left( \sum_{i=1}^{k} \omega_i \left( \tilde{Z}_i\, \tilde{p}_i(x) \right)^{1/\beta} \right)^{\beta}. \tag{30}$$

The mixing policy is therefore:

$$\tilde{p}_{M,F}(s' \mid s) \propto \left( \sum_{i=1}^{k} \omega_i \left( \tilde{Z}_i \tilde{u}_i(s) \cdot \tilde{p}_{i,F}(s' \mid s) \right)^{1/\beta} \right)^{\beta}, \tag{31}$$

**Additional results.** Tables A3 to A9 and Fig. A8 show additional results.

**Ablation: ensemble baseline and training objectives.** Table A3 compares our mixing policy against the ensemble baseline for both SubTB- and TB-trained base GFlowNets. The ensemble baseline mixes forward policies without the reaching probability $u_i(\cdot)$, isolating the contribution of flow-based weighting. Our method consistently outperforms the ensemble across most settings, confirming that the reaching probability is essential. Results are reported with our DB $F$ variant for both SubTB and TB, and with the Model $F$ variant for SubTB.

*Table A3.* Ensemble baseline vs. our mixing policy on molecule generation tasks. Base GFlowNets are trained with either SubTB or TB. We report the average reward and diversity of the top-10 samples across different objective combinations (mean $\pm$ std over 3 seeds). *ALL* denotes all three objectives (SEH/GAP, SA, and QED).

| | | Reward ↑ | | | | | Diversity ↑ | | | | |
| | | SubTB | | | TB | | SubTB | | | TB | |
| | | Ensemble | Ours (Model $F$) | Ours (DB $F$) | Ensemble | Ours (DB $F$) | Ensemble | Ours (Model $F$) | Ours (DB $F$) | Ensemble | Ours (DB $F$) |
|---|---|---|---|---|---|---|---|---|---|---|---|
| Fragment | SEH-SA | $0.818_{\pm0.006}$ | $0.836_{\pm0.001}$ | $0.835_{\pm0.003}$ | $0.836_{\pm0.007}$ | $\mathbf{0.851}_{\pm0.000}$ | $0.548_{\pm0.011}$ | $0.562_{\pm0.005}$ | $0.554_{\pm0.006}$ | $0.537_{\pm0.022}$ | $\mathbf{0.567}_{\pm0.003}$ |
| | SEH-QED | $0.648_{\pm0.005}$ | $0.775_{\pm0.010}$ | $0.777_{\pm0.006}$ | $0.656_{\pm0.004}$ | $\mathbf{0.787}_{\pm0.003}$ | $0.499_{\pm0.006}$ | $0.564_{\pm0.006}$ | $\mathbf{0.565}_{\pm0.013}$ | $0.492_{\pm0.014}$ | $0.560_{\pm0.004}$ |
| | SA-QED | $0.606_{\pm0.003}$ | $0.797_{\pm0.010}$ | $0.790_{\pm0.008}$ | $0.607_{\pm0.005}$ | $\mathbf{0.801}_{\pm0.005}$ | $0.587_{\pm0.007}$ | $\mathbf{0.624}_{\pm0.007}$ | $0.621_{\pm0.012}$ | $0.596_{\pm0.004}$ | $0.617_{\pm0.004}$ |
| | ALL | $0.617_{\pm0.001}$ | $0.741_{\pm0.009}$ | $0.742_{\pm0.010}$ | $0.625_{\pm0.004}$ | $\mathbf{0.743}_{\pm0.005}$ | $0.563_{\pm0.005}$ | $0.608_{\pm0.009}$ | $\mathbf{0.610}_{\pm0.008}$ | $0.553_{\pm0.011}$ | $0.601_{\pm0.006}$ |
| QM9 | GAP-SA | $0.817_{\pm0.009}$ | $0.873_{\pm0.003}$ | $0.868_{\pm0.003}$ | $0.810_{\pm0.007}$ | $\mathbf{0.891}_{\pm0.002}$ | $\mathbf{0.908}_{\pm0.004}$ | $0.899_{\pm0.010}$ | $0.902_{\pm0.010}$ | $0.903_{\pm0.008}$ | $0.883_{\pm0.010}$ |
| | GAP-QED | $0.755_{\pm0.006}$ | $0.778_{\pm0.004}$ | $0.781_{\pm0.005}$ | $0.747_{\pm0.020}$ | $\mathbf{0.782}_{\pm0.005}$ | $\mathbf{0.893}_{\pm0.008}$ | $0.880_{\pm0.014}$ | $0.881_{\pm0.014}$ | $0.885_{\pm0.010}$ | $0.874_{\pm0.008}$ |
| | SA-QED | $0.735_{\pm0.017}$ | $0.710_{\pm0.013}$ | $0.689_{\pm0.014}$ | $\mathbf{0.752}_{\pm0.000}$ | $0.718_{\pm0.007}$ | $0.728_{\pm0.081}$ | $0.778_{\pm0.040}$ | $\mathbf{0.826}_{\pm0.019}$ | $0.643_{\pm0.021}$ | $0.752_{\pm0.014}$ |
| | ALL | $0.668_{\pm0.006}$ | $0.727_{\pm0.005}$ | $0.719_{\pm0.004}$ | $0.649_{\pm0.014}$ | $\mathbf{0.733}_{\pm0.008}$ | $\mathbf{0.899}_{\pm0.004}$ | $0.872_{\pm0.014}$ | $0.882_{\pm0.010}$ | $0.897_{\pm0.013}$ | $0.868_{\pm0.009}$ |

**Multi-objective optimization metrics.** We use three multi-objective optimization metrics. Hypervolume (↑) is the volume dominated by the Pareto front approximation, capturing both convergence and spread. IGD+ (↓) is the average distance from the reference Pareto front to the method's approximation, a Pareto-compliant variant of IGD (Ishibuchi et al., 2015). Pareto Count (↑) is the number of non-dominated solutions found across all generated samples. The hypervolume is reported in the main text (Table 4). IGD+ (Table A4) and Pareto Count (Table A5) are reported below. Our method achieves competitive IGD+ and Pareto Count compared to the baselines, consistent with the hypervolume results.

*Table A4.* IGD+ (↓) on molecule generation tasks. We report mean $\pm$ std over 3 seeds. *ALL* denotes all three objectives (SEH/GAP, SA, and QED).

| | | IGD+ ↓ | | | | |
| | | MOGFN | HN-GFN | OP-GFN | Ours (Model F) | Ours (DB F) |
|---|---|---|---|---|---|---|
| Fragment | SEH-SA | $0.058_{\pm0.002}$ | $\mathbf{0.053}_{\pm0.001}$ | $0.065_{\pm0.008}$ | $0.056_{\pm0.002}$ | $0.060_{\pm0.003}$ |
| | SEH-QED | $0.075_{\pm0.009}$ | $0.082_{\pm0.008}$ | $0.144_{\pm0.012}$ | $\mathbf{0.074}_{\pm0.005}$ | $0.079_{\pm0.009}$ |
| | SA-QED | $0.072_{\pm0.005}$ | $\mathbf{0.069}_{\pm0.005}$ | $0.115_{\pm0.007}$ | $0.070_{\pm0.001}$ | $0.071_{\pm0.006}$ |
| | ALL | $0.111_{\pm0.004}$ | $0.117_{\pm0.002}$ | $0.185_{\pm0.023}$ | $0.110_{\pm0.009}$ | $\mathbf{0.108}_{\pm0.008}$ |
| QM9 | GAP-SA | $0.046_{\pm0.026}$ | $0.036_{\pm0.021}$ | $0.041_{\pm0.018}$ | $\mathbf{0.023}_{\pm0.001}$ | $0.028_{\pm0.003}$ |
| | GAP-QED | $0.140_{\pm0.003}$ | $\mathbf{0.134}_{\pm0.006}$ | $0.136_{\pm0.003}$ | $0.142_{\pm0.001}$ | $0.136_{\pm0.002}$ |
| | SA-QED | $0.153_{\pm0.009}$ | $0.142_{\pm0.001}$ | $0.137_{\pm0.003}$ | $\mathbf{0.133}_{\pm0.003}$ | $0.140_{\pm0.005}$ |
| | ALL | $0.138_{\pm0.004}$ | $0.140_{\pm0.004}$ | $0.164_{\pm0.015}$ | $\mathbf{0.132}_{\pm0.001}$ | $\mathbf{0.132}_{\pm0.002}$ |

*Table A5.* Pareto Count (↑) on molecule generation tasks. We report the number of non-dominated solutions (mean $\pm$ std over 3 seeds). *ALL* denotes all three objectives (SEH/GAP, SA, and QED).

| | | MOGFN | HN-GFN | OP-GFN | Ours (Model F) | Ours (DB F) |
|---|---|---|---|---|---|---|
| Fragment | SEH-SA | $9_{\pm2}$ | $7_{\pm0}$ | $8_{\pm2}$ | $9_{\pm2}$ | $\mathbf{11}_{\pm2}$ |
| | SEH-QED | $16_{\pm2}$ | $11_{\pm1}$ | $11_{\pm4}$ | $14_{\pm2}$ | $\mathbf{18}_{\pm1}$ |
| | SA-QED | $6_{\pm1}$ | $6_{\pm1}$ | $\mathbf{7}_{\pm0}$ | $6_{\pm3}$ | $6_{\pm1}$ |
| | ALL | $\mathbf{119}_{\pm18}$ | $76_{\pm11}$ | $68_{\pm12}$ | $113_{\pm5}$ | $99_{\pm22}$ |
| QM9 | GAP-SA | $6_{\pm3}$ | $9_{\pm2}$ | $7_{\pm1}$ | $\mathbf{15}_{\pm2}$ | $13_{\pm1}$ |
| | GAP-QED | $12_{\pm1}$ | $\mathbf{15}_{\pm2}$ | $9_{\pm3}$ | $13_{\pm2}$ | $10_{\pm2}$ |
| | SA-QED | $8_{\pm2}$ | $10_{\pm3}$ | $10_{\pm2}$ | $\mathbf{12}_{\pm1}$ | $11_{\pm2}$ |
| | ALL | $77_{\pm20}$ | $65_{\pm25}$ | $57_{\pm19}$ | $\mathbf{101}_{\pm16}$ | $89_{\pm16}$ |

*Table A6.* Results of logical operators with our mixing policy (Model $F$) on molecule generation tasks (mean over 3 seeds). We report the percentage of valid samples falling into each of 8 bins based on reward thresholds: Fragment-based (SEH: 0.5, SA: 0.6, QED: 0.25) and Atom-based QM9 (GAP: 0.85, SA: 0.3, QED: 0.3). **Bold** indicates the *target bin*: for harmonic mean ($\otimes$), where all objectives are high; for contrast (◑), where the first objective is high and the rest low.

| | Fragment-based | | | | | | | | | Atom-based (QM9) | | | | | | | |
| --- | --- | --- | --- | --- | --- | --- | --- | --- | --- | --- | --- | --- | --- | --- | --- | --- |
| SEH | low | | | | high | | | | GAP | low | | | | high | | | |
| SA | low | | high | | low | | high | | SA | low | | high | | low | | high | |
| QED | low | high | low | high | low | high | low | high | QED | low | high | low | high | low | high | low | high |
| *Base distributions* | | | | | | | | | | | | | | | | | |
| $p_{\mathrm{SEH}}$ | 1 | 0 | 0 | 0 | 56 | 12 | 28 | 3 | $p_{\mathrm{GAP}}$ | 0 | 0 | 1 | 0 | 43 | 11 | 35 | 10 |
| $p_{\mathrm{SA}}$ | 0 | 0 | 74 | 5 | 0 | 0 | 18 | 3 | $p_{\mathrm{SA}}$ | 0 | 0 | 3 | 32 | 0 | 0 | 9 | 56 |
| $p_{\mathrm{QED}}$ | 0 | 38 | 0 | 25 | 0 | 24 | 0 | 13 | $p_{\mathrm{QED}}$ | 0 | 1 | 0 | 1 | 2 | 54 | 2 | 40 |
| *Harmonic mean* ($\otimes$) | | | | | | | | | | | | | | | | | |
| $p_{\mathrm{SEH}} \otimes p_{\mathrm{SA}}$ | 1 | 0 | 7 | 0 | 3 | 1 | 77 | 11 | $p_{\mathrm{GAP}} \otimes p_{\mathrm{SA}}$ | 0 | 0 | 1 | 1 | 4 | 7 | 30 | **57** |
| $p_{\mathrm{SEH}} \otimes p_{\mathrm{QED}}$ | 0 | 9 | 0 | 4 | 0 | 61 | 0 | 26 | $p_{\mathrm{GAP}} \otimes p_{\mathrm{QED}}$ | 0 | 0 | 0 | 0 | 13 | 46 | 8 | **33** |
| $p_{\mathrm{SA}} \otimes p_{\mathrm{QED}}$ | 0 | 5 | 0 | 68 | 0 | 3 | 0 | 24 | $p_{\mathrm{SA}} \otimes p_{\mathrm{QED}}$ | 0 | 1 | 0 | 11 | 2 | 11 | 5 | **70** |
| $p_{\mathrm{SEH}} \otimes p_{\mathrm{SA}} \otimes p_{\mathrm{QED}}$ | 0 | 6 | 0 | 12 | 0 | 17 | 0 | 65 | $p_{\mathrm{GAP}} \otimes p_{\mathrm{SA}} \otimes p_{\mathrm{QED}}$ | 3 | 0 | 0 | 2 | 6 | 17 | 11 | **61** |
| *Contrast* (◑) | | | | | | | | | | | | | | | | | |
| $p_{\mathrm{SEH}}$ ◑ $p_{\mathrm{SA}}$ | 1 | 0 | 0 | 0 | **62** | 14 | 20 | 3 | $p_{\mathrm{GAP}}$ ◑ $p_{\mathrm{SA}}$ | 1 | 0 | 1 | 0 | **47** | 8 | 37 | 6 |
| $p_{\mathrm{SA}}$ ◑ $p_{\mathrm{SEH}}$ | 0 | 0 | **80** | 5 | 0 | 0 | 13 | 2 | $p_{\mathrm{SA}}$ ◑ $p_{\mathrm{GAP}}$ | 0 | 0 | **1** | 35 | 0 | 0 | 7 | 57 |
| $p_{\mathrm{SEH}}$ ◑ $p_{\mathrm{QED}}$ | 1 | 0 | 1 | 0 | **57** | 10 | 28 | 3 | $p_{\mathrm{GAP}}$ ◑ $p_{\mathrm{QED}}$ | 0 | 0 | 1 | 0 | **47** | 7 | 38 | 7 |
| $p_{\mathrm{QED}}$ ◑ $p_{\mathrm{SEH}}$ | 0 | **40** | 0 | 26 | 0 | 22 | 0 | 12 | $p_{\mathrm{QED}}$ ◑ $p_{\mathrm{GAP}}$ | 0 | **1** | 0 | 1 | 1 | 56 | 1 | 40 |
| $p_{\mathrm{SA}}$ ◑ $p_{\mathrm{QED}}$ | 1 | 0 | **77** | 4 | 0 | 0 | 16 | 2 | $p_{\mathrm{SA}}$ ◑ $p_{\mathrm{QED}}$ | 0 | 0 | **4** | 30 | 0 | 0 | 13 | 53 |
| $p_{\mathrm{QED}}$ ◑ $p_{\mathrm{SA}}$ | 0 | **43** | 0 | 19 | 0 | 27 | 0 | 11 | $p_{\mathrm{QED}}$ ◑ $p_{\mathrm{SA}}$ | 0 | **0** | 0 | 1 | 2 | 56 | 1 | 40 |
| $p_{\mathrm{SEH}}$ ◑ $p_{\mathrm{SA}}$ ◑ $p_{\mathrm{QED}}$ | 1 | 0 | 0 | 0 | **64** | 12 | 20 | 3 | $p_{\mathrm{GAP}}$ ◑ $p_{\mathrm{SA}}$ ◑ $p_{\mathrm{QED}}$ | 1 | 0 | 1 | 0 | **49** | 5 | 39 | 5 |
| $p_{\mathrm{SA}}$ ◑ $p_{\mathrm{SEH}}$ ◑ $p_{\mathrm{QED}}$ | 0 | 0 | **82** | 4 | 0 | 0 | 12 | 2 | $p_{\mathrm{SA}}$ ◑ $p_{\mathrm{GAP}}$ ◑ $p_{\mathrm{QED}}$ | 0 | 0 | **3** | 34 | 0 | 0 | 11 | 52 |
| $p_{\mathrm{QED}}$ ◑ $p_{\mathrm{SEH}}$ ◑ $p_{\mathrm{SA}}$ | 0 | **45** | 0 | 20 | 0 | 25 | 0 | 10 | $p_{\mathrm{QED}}$ ◑ $p_{\mathrm{GAP}}$ ◑ $p_{\mathrm{SA}}$ | 2 | **0** | 0 | 1 | 1 | 55 | 1 | 40 |

*Table A7.* Results of logical operators with our mixing policy (DB $F$) on molecule generation tasks (mean over 3 seeds). We report the percentage of valid samples falling into each of 8 bins based on reward thresholds: Fragment-based (SEH: 0.5, SA: 0.6, QED: 0.25) and Atom-based QM9 (GAP: 0.85, SA: 0.3, QED: 0.3). **Bold** indicates the *target bin*: for harmonic mean ($\otimes$), where all objectives are high; for contrast (◑), where the first objective is high and the rest low.

| | Fragment-based | | | | | | | | | Atom-based (QM9) | | | | | | | |
| --- | --- | --- | --- | --- | --- | --- | --- | --- | --- | --- | --- | --- | --- | --- | --- | --- | --- |
| SEH | low | | | | high | | | | GAP | low | | | | high | | | |
| SA | low | | high | | low | | high | | SA | low | | high | | low | | high | |
| QED | low | high | low | high | low | high | low | high | QED | low | high | low | high | low | high | low | high |
| *Base distributions* | | | | | | | | | | | | | | | | | |
| $p_{\mathrm{SEH}}$ | 1 | 0 | 0 | 0 | 56 | 12 | 28 | 3 | $p_{\mathrm{GAP}}$ | 0 | 0 | 1 | 0 | 43 | 11 | 35 | 10 |
| $p_{\mathrm{SA}}$ | 0 | 0 | 74 | 5 | 0 | 0 | 18 | 3 | $p_{\mathrm{SA}}$ | 0 | 0 | 3 | 32 | 0 | 0 | 9 | 56 |
| $p_{\mathrm{QED}}$ | 0 | 38 | 0 | 25 | 0 | 24 | 0 | 13 | $p_{\mathrm{QED}}$ | 0 | 1 | 0 | 1 | 2 | 54 | 2 | 40 |
| *Harmonic mean* ($\otimes$) | | | | | | | | | | | | | | | | | |
| $p_{\mathrm{SEH}} \otimes p_{\mathrm{SA}}$ | 1 | 0 | 6 | 1 | 3 | 1 | 78 | 10 | $p_{\mathrm{GAP}} \otimes p_{\mathrm{SA}}$ | 0 | 0 | 1 | 1 | 5 | 7 | 30 | **56** |
| $p_{\mathrm{SEH}} \otimes p_{\mathrm{QED}}$ | 0 | 8 | 0 | 3 | 0 | 64 | 0 | 25 | $p_{\mathrm{GAP}} \otimes p_{\mathrm{QED}}$ | 0 | 0 | 0 | 0 | 12 | 44 | 9 | **35** |
| $p_{\mathrm{SA}} \otimes p_{\mathrm{QED}}$ | 0 | 5 | 0 | 68 | 0 | 3 | 0 | 24 | $p_{\mathrm{SA}} \otimes p_{\mathrm{QED}}$ | 0 | 0 | 0 | 12 | 1 | 9 | 5 | **73** |
| $p_{\mathrm{SEH}} \otimes p_{\mathrm{SA}} \otimes p_{\mathrm{QED}}$ | 0 | 5 | 0 | 10 | 0 | 18 | 1 | 66 | $p_{\mathrm{GAP}} \otimes p_{\mathrm{SA}} \otimes p_{\mathrm{QED}}$ | 2 | 0 | 0 | 2 | 4 | 17 | 12 | **63** |
| *Contrast* (◑) | | | | | | | | | | | | | | | | | |
| $p_{\mathrm{SEH}}$ ◑ $p_{\mathrm{SA}}$ | 1 | 0 | 0 | 0 | **62** | 14 | 20 | 3 | $p_{\mathrm{GAP}}$ ◑ $p_{\mathrm{SA}}$ | 1 | 0 | 1 | 0 | **48** | 7 | 36 | 7 |
| $p_{\mathrm{SA}}$ ◑ $p_{\mathrm{SEH}}$ | 0 | 0 | **80** | 5 | 0 | 0 | 13 | 2 | $p_{\mathrm{SA}}$ ◑ $p_{\mathrm{GAP}}$ | 0 | 0 | **2** | 35 | 0 | 0 | 7 | 56 |
| $p_{\mathrm{SEH}}$ ◑ $p_{\mathrm{QED}}$ | 1 | 0 | 1 | 0 | **57** | 10 | 28 | 3 | $p_{\mathrm{GAP}}$ ◑ $p_{\mathrm{QED}}$ | 0 | 0 | 1 | 0 | **46** | 8 | 38 | 7 |
| $p_{\mathrm{QED}}$ ◑ $p_{\mathrm{SEH}}$ | 0 | **40** | 0 | 26 | 0 | 22 | 0 | 12 | $p_{\mathrm{QED}}$ ◑ $p_{\mathrm{GAP}}$ | 0 | **1** | 0 | 1 | 1 | 55 | 1 | 41 |
| $p_{\mathrm{SA}}$ ◑ $p_{\mathrm{QED}}$ | 0 | 0 | **78** | 4 | 0 | 0 | 16 | 2 | $p_{\mathrm{SA}}$ ◑ $p_{\mathrm{QED}}$ | 0 | 0 | **4** | 30 | 0 | 0 | 12 | 54 |
| $p_{\mathrm{QED}}$ ◑ $p_{\mathrm{SA}}$ | 0 | **43** | 0 | 19 | 0 | 27 | 0 | 11 | $p_{\mathrm{QED}}$ ◑ $p_{\mathrm{SA}}$ | 0 | **1** | 0 | 1 | 1 | 56 | 1 | 40 |
| $p_{\mathrm{SEH}}$ ◑ $p_{\mathrm{SA}}$ ◑ $p_{\mathrm{QED}}$ | 1 | 0 | 0 | 0 | **63** | 12 | 21 | 3 | $p_{\mathrm{GAP}}$ ◑ $p_{\mathrm{SA}}$ ◑ $p_{\mathrm{QED}}$ | 1 | 0 | 1 | 0 | **50** | 6 | 37 | 5 |
| $p_{\mathrm{SA}}$ ◑ $p_{\mathrm{SEH}}$ ◑ $p_{\mathrm{QED}}$ | 0 | 0 | **82** | 4 | 0 | 0 | 13 | 1 | $p_{\mathrm{SA}}$ ◑ $p_{\mathrm{GAP}}$ ◑ $p_{\mathrm{QED}}$ | 0 | 0 | **3** | 33 | 0 | 0 | 10 | 54 |
| $p_{\mathrm{QED}}$ ◑ $p_{\mathrm{SEH}}$ ◑ $p_{\mathrm{SA}}$ | 0 | **45** | 0 | 19 | 0 | 26 | 0 | 10 | $p_{\mathrm{QED}}$ ◑ $p_{\mathrm{GAP}}$ ◑ $p_{\mathrm{SA}}$ | 1 | **0** | 0 | 1 | 1 | 56 | 1 | 40 |

*Table A8.* Results of logical operators with classifier-guided baseline on molecule generation tasks (mean over 3 seeds). We report the percentage of valid samples falling into each of 8 bins based on reward thresholds: Fragment-based (SEH: 0.5, SA: 0.6, QED: 0.25) and Atom-based QM9 (GAP: 0.85, SA: 0.3, QED: 0.3). **Bold** indicates the *target bin*: for harmonic mean (⊗), where all objectives are high; for contrast (◖), where the first objective is high and the rest low.

| SEH | low | low | low | low | high | high | high | high | GAP | low | low | low | low | high | high | high | high |
|---|---|---|---|---|---|---|---|---|---|---|---|---|---|---|---|---|---|
| **SA** | low | low | high | high | low | low | high | high | **SA** | low | low | high | high | low | low | high | high |
| **QED** | low | high | low | high | low | high | low | high | **QED** | low | high | low | high | low | high | low | high |
| *Base distributions* | | | | | | | | | | | | | | | | | |
| $p_{\text{SEH}}$ | 1 | 0 | 0 | 0 | **56** | **12** | **28** | **3** | $p_{\text{GAP}}$ | 0 | 0 | 1 | 0 | **43** | **11** | **35** | **10** |
| $p_{\text{SA}}$ | 0 | 0 | **74** | **5** | 0 | 0 | **18** | **3** | $p_{\text{SA}}$ | 0 | 0 | **3** | **32** | 0 | 0 | **9** | **56** |
| $p_{\text{QED}}$ | 0 | **38** | 0 | **25** | 0 | **24** | 0 | **13** | $p_{\text{QED}}$ | 0 | **1** | 0 | **1** | 2 | **54** | 2 | **40** |
| *Harmonic mean (⊗)* | | | | | | | | | | | | | | | | | |
| $p_{\text{SEH}} \otimes p_{\text{SA}}$ | 1 | 0 | 9 | 1 | 7 | 1 | **72** | **9** | $p_{\text{GAP}} \otimes p_{\text{SA}}$ | 0 | 0 | 4 | 4 | 9 | 5 | **39** | **39** |
| $p_{\text{SEH}} \otimes p_{\text{QED}}$ | 0 | 12 | 0 | 6 | 1 | **52** | 1 | **28** | $p_{\text{GAP}} \otimes p_{\text{QED}}$ | 0 | 0 | 0 | 1 | 13 | **41** | 10 | **35** |
| $p_{\text{SA}} \otimes p_{\text{QED}}$ | 0 | 6 | 2 | **58** | 0 | 3 | 1 | **30** | $p_{\text{SA}} \otimes p_{\text{QED}}$ | 0 | 0 | 1 | **21** | 1 | 5 | 7 | **65** |
| $p_{\text{SEH}} \otimes p_{\text{SA}} \otimes p_{\text{QED}}$ | 1 | 1 | 6 | 7 | 2 | 5 | 38 | **40** | $p_{\text{GAP}} \otimes p_{\text{SA}} \otimes p_{\text{QED}}$ | 0 | 0 | 1 | 6 | 8 | 19 | 15 | **51** |
| *Contrast (◖)* | | | | | | | | | | | | | | | | | |
| $p_{\text{SEH}} ◖ p_{\text{SA}}$ | 1 | 0 | 1 | 0 | **56** | **12** | 27 | 3 | $p_{\text{GAP}} ◖ p_{\text{SA}}$ | 1 | 0 | 0 | 0 | **44** | **11** | 34 | 10 |
| $p_{\text{SA}} ◖ p_{\text{SEH}}$ | 0 | 0 | **75** | **5** | 0 | 0 | 17 | 3 | $p_{\text{SA}} ◖ p_{\text{GAP}}$ | 0 | 0 | **2** | **32** | 0 | 0 | 9 | 57 |
| $p_{\text{SEH}} ◖ p_{\text{QED}}$ | 1 | 0 | 1 | 0 | **57** | 12 | **26** | 3 | $p_{\text{GAP}} ◖ p_{\text{QED}}$ | 0 | 0 | 1 | 0 | **45** | 11 | **34** | 9 |
| $p_{\text{QED}} ◖ p_{\text{SEH}}$ | 0 | **38** | 0 | **25** | 0 | 24 | 0 | 13 | $p_{\text{QED}} ◖ p_{\text{GAP}}$ | 0 | **1** | 0 | **1** | 2 | 54 | 1 | 41 |
| $p_{\text{SA}} ◖ p_{\text{QED}}$ | 0 | 0 | **75** | 5 | 0 | 0 | **17** | 3 | $p_{\text{SA}} ◖ p_{\text{QED}}$ | 0 | 0 | **2** | 32 | 0 | 0 | **10** | 56 |
| $p_{\text{QED}} ◖ p_{\text{SA}}$ | 0 | **38** | 0 | 25 | 0 | **24** | 0 | 13 | $p_{\text{QED}} ◖ p_{\text{SA}}$ | 0 | **0** | 0 | 1 | 2 | **54** | 2 | 41 |
| $p_{\text{SEH}} ◖ p_{\text{SA}} ◖ p_{\text{QED}}$ | 2 | 0 | 1 | 0 | **57** | 11 | 26 | 3 | $p_{\text{GAP}} ◖ p_{\text{SA}} ◖ p_{\text{QED}}$ | 0 | 0 | 0 | 0 | **45** | 10 | 36 | 9 |
| $p_{\text{SA}} ◖ p_{\text{SEH}} ◖ p_{\text{QED}}$ | 0 | 0 | **74** | 5 | 0 | 0 | 18 | 3 | $p_{\text{SA}} ◖ p_{\text{GAP}} ◖ p_{\text{QED}}$ | 0 | 0 | **2** | 33 | 0 | 0 | 9 | 56 |
| $p_{\text{QED}} ◖ p_{\text{SEH}} ◖ p_{\text{SA}}$ | 0 | **38** | 0 | 24 | 0 | 24 | 0 | 14 | $p_{\text{QED}} ◖ p_{\text{GAP}} ◖ p_{\text{SA}}$ | 0 | **0** | 0 | 1 | 2 | 55 | 1 | 41 |

**Effect of reward correlations on logical operators.** The effectiveness of the contrast operator depends on the correlation structure of the underlying reward distributions. Table A9 illustrates challenging cases: contrast operations such as $p_{\text{QED}} ◖ p_{\text{GAP}}$ and $p_{\text{SA}} ◖ p_{\text{QED}}$ show limited separation. As shown in Fig. A8, we observe that among samples generated by the trained GFlowNets, those with high QED tend to also have high GAP, and those with high SA tend to have high QED. When rewards exhibit such positive correlations in the generated samples, the contrast operator struggles to isolate the target region since samples that score high on one objective inherently tend to score high on the other.

*Table A9.* Challenging cases of the contrast operator on atom-based molecule generation (QM9). We report the percentage of samples falling into each of 8 bins based on reward thresholds (GAP: 0.85, SA: 0.3, QED: 0.3). **Bold** indicates the *target bin*: for contrast (◖), where only the first objective is high.

| GAP | low | low | low | low | high | high | high | high |
|---|---|---|---|---|---|---|---|---|
| **SA** | low | low | high | high | low | low | high | high |
| **QED** | low | high | low | high | low | high | low | high |
| $p_{\text{GAP}}$ | 0 | 0 | 1 | 0 | **43** | **11** | **35** | **10** |
| $p_{\text{SA}}$ | 0 | 0 | **3** | **32** | 0 | 0 | **9** | **56** |
| $p_{\text{QED}}$ | 0 | **1** | 0 | **1** | 2 | **54** | 2 | **40** |
| $p_{\text{SA}} ◖ p_{\text{QED}}$ | 0 | 0 | **4** | 30 | 0 | 0 | **13** | 53 |
| $p_{\text{QED}} ◖ p_{\text{GAP}}$ | 0 | **1** | 0 | **1** | 1 | 56 | 1 | 40 |
| $p_{\text{SA}} ◖ p_{\text{GAP}} ◖ p_{\text{QED}}$ | 0 | 0 | **3** | 34 | 0 | 0 | 11 | 52 |
| $p_{\text{QED}} ◖ p_{\text{GAP}} ◖ p_{\text{SA}}$ | 2 | **0** | 0 | 1 | 1 | 55 | 1 | 40 |

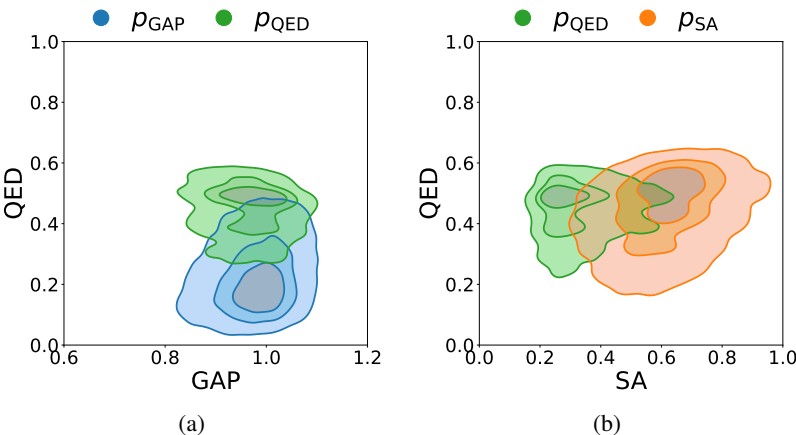

(a)                    (b)

*Figure A8.* Reward distributions of base models in QM9, illustrating positive correlations between objectives. We observe that (a) high QED samples tend to have high GAP, and (b) high SA samples tend to have high QED.

