# OpenReview forum: "Routing by Reaching: Composition of Pre-trained GFlowNets for Multi-Objective Generation"
_ICML.cc/2026/Conference — ICML 2026 regular_

### Official Review · Reviewer_9JbA · 2026-03-09

**Soundness:** 2
**Presentation:** 3
**Significance:** 2
**Originality:** 3
**Overall Recommendation:** 4
**Confidence:** 4

**Summary:**

This paper addresses the problem of multi-objective generation with GFlowNets, where the goal is to sample from a distribution defined over combinations of multiple reward functions. Prior approaches either preference-conditioned GFlowNets (MOGFN, HN-GFN) or compositional sculpting all require additional training whenever a new objective set is introduced, which is computationally expensive and inflexible. The central contribution is a training-free mixing policy that composes pre-trained single-objective GFlowNets at inference time by weighting each model's transition probabilities by its reaching probability. The paper further provides a theoretical exactness guarantee for linear scalarization (\beta=1), and characterizes the approximation error for non-linear operators via a distortion factor, showing empirically that deviations are small in high-reward regions where it matters. Experiments on a 2D grid and real-world molecule generation (fragment- and atom-based) demonstrate that the method matches or outperforms training-required baselines on both sample quality and composition accuracy, while being faster at inference than classifier-guided composition.

**Compliance With Llm Reviewing Policy:**

Affirmed.

**Final Justification:**

The paper proposes a training-free mixing policy for composing pre-trained GFlowNets at inference time, with an exact recovery guarantee for linear scalarization (β=1) and empirical validation for non-linear operators. My original concerns centered on four issues: the theory-practice gap (β=1 vs β=32/64 in experiments), TB incompatibility, diversity degradation, and sensitivity to base model quality.
The rebuttal addressed all four. Most notably, the authors proposed an on-the-fly reaching probability estimator that makes the method fully compatible with TB-trained models, resolving what I considered the most significant practical limitation. The β>1 distortion analysis on the 2D grid, multi-seed results with standard deviations, and the base model convergence ablation collectively strengthened the empirical foundation. The diversity gap was shown to be modest in most settings.
The core idea of using reaching probabilities as mixing weights is intuitive and theoretically grounded, and the unified treatment of scalarization and logical operators within a single framework is a meaningful contribution. While the theoretical guarantee remains restricted to β=1, the practical performance is competitive with training-required baselines across both synthetic and molecular benchmarks. I raise my score from 3 to 4 (Weak Accept), contingent on the revised manuscript incorporating the OTF estimator in the main text and qualifying the exactness claim.

**Key Questions For Authors:**

Q1. The mixing policy inherits directly from pre-trained base models without correction. How does the quality of individual base GFlowNets affect the composed distribution? Does error compound across objectives? An ablation over base model convergence quality would clarify practical robustness.

Q2. Exactness is proven only for $\beta=1$, yet all molecule generation experiments use $\beta=32/64$. Can the authors provide either a formal error bound or an empirical distortion analysis (analogous to Figure 3) for the reward-sharpened composition in Appendix B.2.1? Without this, the core experimental setting lacks theoretical grounding.

Q3. Table 3 shows diversity consistently below baselines. Is this an inherent consequence of mixing mode-biased base policies — i.e., the composed distribution concentrating on the intersection of each base model's high-reward modes? If so, under what conditions is this limitation most severe?

Q4.The method excludes Trajectory Balance, which is widely used in recent GFlowNet work. Is approximate reaching probability estimation (e.g., via Monte Carlo rollouts) feasible for TB-trained models, and what would the performance cost be?

**Limitations:**

The limitations discussion is insufficiently thorough relative to the technical weaknesses present in the paper. Specifically, the following points deserve explicit acknowledgment.
- TB incompatibility is not discussed as a limitation in the main text at all, despite being a meaningful constraint on applicability.
- The $\beta≠1$ approximation gap, the fact that the core theoretical guarantee does not apply to the primary experimental setting, is not framed as a limitation anywhere in the paper.
- Diversity degradation is visible in Table 3 but never discussed critically as a potential failure mode of the method.

**Strengths And Weaknesses:**

Strengths

- The use of reaching probability as mixing weights is intuitive and novel
- Exact recovery guarantee for linear scalarization ($\beta=1$)
- Inference speedup over classifier-guided composition; near-perfect molecular validity preserved
- Unified framework covering both scalarization and logical operators

Weaknesses

- The exactness guarantee holds only at $\beta=1$, yet all real-world experiments use $\beta=32/64$. The reward-sharpened composition in Appendix B.2.1 bridges this gap without theoretical grounding or systematic empirical validation. The paper's primary experimental setting is thus not covered by its main theoretical result that a fundamental gap that requires resolution before the method can be reliably built upon.
- TB incompatibility is a significant practical constraint given its widespread use in recent GFlowNet work, yet it is only briefly mentioned in the appendix. This is a transparency issue for practitioners evaluating adoption.
- Diversity is consistently lower than or comparable to baselines in Table 3, with no critical analysis offered. This may reflect a structural limitation: mixing mode-biased base policies likely concentrates the composed distribution on the intersection of high-reward regions, reducing effective support. For a method targeting multi-objective drug discovery, where diverse exploration is the core value of GFlowNets, this warrants explicit discussion.

---

> ### Author Rebuttal · Authors · 2026-03-31
>
> We sincerely appreciate your thoughtful comments and constructive suggestions. We address each of your concerns in the following response. For supplementary tables (Table R#) and figures (Figure R#) referenced throughout our responses, please refer to the following PDF: https://anonymous.4open.science/r/ICML2026_Rebuttal-CD17/icml2026-rebuttal.pdf.
>
> ---
>
> **W1&Q2.** Exactness holds only at $\beta=1$, but experiments use $\beta \gg 1$. Distortion analysis for $\beta>1$?
>
> **A1.** We acknowledge that our exactness guarantee formally holds only at $\beta=1$, and we agree that bridging this gap for $\beta \neq 1$ is essential.
>
> **Why empirical distortion analysis is intractable for molecules.** Even an empirical approximation of the distortion factor is difficult in the molecular setting: (1) the reaching probability $u_M(x)$ cannot be reliably estimated for a given terminal state, since there are exponentially many trajectories leading to it and sampling captures only one, (2) the molecular space is far too large to achieve meaningful coverage through sampling, and (3) the learned GFlowNet concentrates samples in high-reward regions, making it impossible to characterize distortion in low-reward regions where deviations are expected to be largest.
>
> **Hypergrid analysis as a proxy (Figure R1).** To provide the requested analysis in a controlled setting where ground truth is available, we conducted experiments on the 2D hypergrid with $\beta \in \{1,2,4\}$. As expected, the global $L_1$ distance increases with $\beta$ ($L_1=0.0028$ for $\beta=1$, $L_1=0.0946$ for $\beta=2$, $L_1=0.1843$ for $\beta=4$). Importantly, the distortion factor analysis shows that in high-reward regions, where $\mathcal{G}(p_1(x),\dots,p_k(x))$ is large, the distortion values cluster tightly around the ideal value $1/Z_M$. The qualitative comparison also confirms that the structural patterns of the target distribution are faithfully preserved in high-reward regions.
>
> **Implications for molecular generation.** This insight transfers to the molecular setting: what matters for practical applications is not the global $L_1$ distance, but whether the method generates high-quality, diverse samples in high-reward regions. As shown in Tables R1, R2, R3, and R4, our method consistently outperforms baselines on both reward/diversity and MOO metrics (Hypervolume, IGD+, Pareto Count) across all settings, confirming that the approximation quality in the practically relevant high-reward regions is sufficient for effective multi-objective molecular generation.
>
> ---
>
> **W2&Q4.** TB incompatibility.
>
> **A2.** We have confirmed that our mixing policy is fully compatible with TB-trained models via a simple on-the-fly (OTF) reaching probability estimator (see Reviewer rSV3 A2 for details). The OTF estimator requires only $p_F$ and $p_B$, which TB already provides, and faithfully reproduces the Model F results. Moreover, TB-trained base models combined with OTF achieve the best reward across all settings (see Table R2).
>
>
> ---
>
> **W3&Q3.** Diversity consistently lower than baselines.
>
> **A3.** With our updated multi-seed results (Table R1), we provide the diversity gap analysis on Fragment (Table R7). The gaps are small in most settings (0.006\~0.011), with the exception of SA-QED (0.033). On QM9, our SubTB Model F variant actually **exceeds** baselines in GAP-QED (0.880 vs 0.865 for HN-GFN), showing that lower diversity is not inherent to our approach.
>
> The modest diversity reduction is a natural reward-diversity trade-off: our mixing policy concentrates samples on the high-reward intersection of base distributions, yielding higher reward at the cost of slightly reduced spread.
>
> ---
>
> **Q1.** How does base model quality affect composed distribution? Does error compound?
>
> **A4.** We ablated base model convergence quality on the 2D hypergrid (Shubert-Diagonal, $\beta=1$) using checkpoints at 5k, 10k, and 20k training steps (Table R9). As base models improve, the composed $L_1$ consistently decreases. This confirms that better-trained base models lead to better mixing results.

---

> > ### Author Rebuttal · Reviewer_9JbA · 2026-04-03
> >
> > I thank the authors for the thorough rebuttal. All oncerns have been addressed. I expect the revised manuscript to incorporate the key additions from the rebuttal, particularly the OTF estimator and the β=1 qualification.

---

> > > ### Author Response · Authors · 2026-04-03
> > >
> > > We sincerely thank the reviewer for the constructive feedback throughout the review process. We confirm that the revised manuscript will incorporate the OTF reaching probability computation (Section 4.1 / Appendix A.2) and qualify the exactness guarantee with the $\beta=1$ condition, as promised in our rebuttal.

---

### Official Review · Reviewer_ZFv8 · 2026-03-11

**Soundness:** 2
**Presentation:** 3
**Significance:** 3
**Originality:** 3
**Overall Recommendation:** 4
**Confidence:** 4

**Summary:**

This paper addresses multi-objective generation with GFlowNets, where the goal is to sample from distributions defined by combinations of multiple reward functions without retraining a model for each new combination. The authors propose a training-free framework that composes pre-trained GFlowNets at inference time by mixing their forward policies, with each model's contribution weighted by its learned reaching probability. For linear scalarization with temperature $\beta=1$, the paper proves that this mixing rule exactly induces the target distribution. For non-linear operators (harmonic mean and contrast), the paper provides an empirical approximation analysis via the distortion factor $\delta(x)$. Experiments on a 32×32 2D grid and two molecular generation benchmarks compare the method against preference-conditioned baselines (MOGFN, HN-GFN) and classifier-guided composition (Garipov et al., 2023).

**Compliance With Llm Reviewing Policy:**

Affirmed.

**Final Justification:**

This paper proposes a training-free framework for composing pre-trained GFlowNets by mixing forward policies weighted by learned reaching probabilities. The framework unifies scalarization and logical composition within a single inference-time procedure, with a provable exactness guarantee for linear scalarization and empirical distortion characterization for non-linear operators.

The theoretical foundation is solid and the practical benefits are well-demonstrated: the 40–70× inference speedup over classifier-guided mixing and the substantial improvement in molecular validity (Table 6) are meaningful results. The unification of two previously separate lines of work represents a genuine conceptual contribution.

The rebuttal substantially changed my evaluation. My three main concerns — absence of distortion analysis for Eq. (26), TB incompatibility, and exclusion of RGFN — were all addressed with technically grounded arguments. In particular, the clarification that Figure R1 directly characterizes the approximation quality of the reward-sharpened scheme Eq. (26), and that OTF computes u(s) exactly under standard convergence assumptions, resolved what I had considered the two most substantive theoretical gaps. The multi-seed experiments and inclusion of the ensemble baseline in molecular experiments further strengthen the empirical claims.

I recommend weak acceptance.

**Key Questions For Authors:**

1. OP-GFN is cited in Section 2 as a related scalarization method and evaluates on the same fragment-based molecular design benchmark used in Table 3. Why is it excluded from all experimental tables? If a direct comparison is feasible, can the authors provide it? Similarly, RGFN uses the sEH reward on the fragment-based benchmark — the same setup as Section 6.1 — yet does not appear in Table 3 or 4. What is the proposed method's performance relative to RGFN on this benchmark?

2. Section 4.2 and Appendix B.2.1 introduce Eq. (26) for mixing reward-sharpened base models at $\beta\neq 1$, but no distortion factor analysis is provided for this case. On the 2D grid (where ground truth is computable), what does $\delta(x)$ look like for $\beta=32/64$ under linear scalarization? Does approximation quality degrade monotonically with $\beta$, or remain stable?

3. Table 3 reports that on QM9 SA-QED scalarization, the proposed method achieves reward 0.729 versus HN-GFN's 0.531 — a gap of nearly 0.2. Over how many random seeds was this reported, and what is the standard deviation? Is this result consistent across seeds, or does it reflect variance in the HN-GFN baseline?

4. The ensemble baseline (uniform mixing without reaching probability weighting) is included in synthetic experiments (Tables 1 and 2) but omitted from all real-world experiments (Tables 3 and 4). Why? Including it in the molecular experiments would confirm that reaching-probability weighting remains essential beyond the synthetic setting, directly strengthening the paper's central claim about the role of state flows.

5. Table A6 shows that the contrast operator fails when objectives are positively correlated (e.g., pQED vs. pGAP in QM9). Does classifier-guided composition also fail in these cases, or does it succeed where the proposed method does not? A comparison on the challenging cases in Table A6 would clarify whether this is a fundamental limitation of the operator or a shortcoming specific to the approximation.

**Limitations:**

Yes

**Strengths And Weaknesses:**

**Strengths**

- The method is genuinely training-free and applies uniformly to both scalarization and logical operators within a single framework, whereas prior work handles these two settings with separate methods. The inference speedup over classifier-guided mixing (40–70× on molecular tasks per Table 5) is a practically meaningful benefit, as is the substantial improvement in generated molecule validity shown in Table 6.

- The distortion factor analysis in Section 5.3 and Figure 3 is a principled effort to characterize when the non-linear approximation is accurate.

---

**Weaknesses**

1. Baseline and related works coverage is insufficient for an ICML 2026 submission.
All scalarization baselines in Tables 1 and 3 (MOGFN, HN-GFN) were published at ICML 2023 and NeurIPS 2023 respectively. At least two directly relevant methods[1,2] published afterward are missing and require comparison or principled justification for exclusion.

[1] Chen Y, Mauch L. Order-Preserving GFlowNets[C]//The Twelfth International Conference on Learning Representations.

[2] Koziarski M, Rekesh A, Shevchuk D, et al. Rgfn: Synthesizable molecular generation using gflownets[J]. Advances in Neural Information Processing Systems, 2024, 37: 46908-46955.

2. The theoretical guarantee applies to $\beta=1$ but all real-world experiments use $\beta=32/64$.
The $\beta\neq1$ scalarization case is handled by Eq. (26) in Appendix B.2.1, which mixes reward-sharpened base models, but no distortion factor analysis is provided for this case analogous to the analysis in Section 5.3 and Figure 3 for logical operators. The paper therefore provides strong theoretical support for a regime ($\beta=1$) that does not correspond to its experimental setting, and only empirical support — without the $\delta(x)$ characterization — for the regime ($\beta>>1$) that does. The 2D grid setting, where ground truth is computable, would permit this analysis at negligible cost.

3. Real-world results in Tables 3 and 4 are reported without standard deviations, and the number of random seeds is not stated.

4. The abstract states the method "exactly recovers the target distribution for linear scalarization" without mentioning the $\beta=1$ restriction. Given that $\beta=1$ is not the operational default in molecular generation experiments, this phrasing is misleading and should be qualified.

5. The method requires base GFlowNets to be trained with objectives that explicitly parameterize the state flow $F(s)$  because the reaching probability estimator $u(s) = F(s)/Z$ relies on learned state flows. This constraint is stated only in Appendix A.2. Trajectory Balance, which does not parameterize $F(s)$, is widely used; the main paper should state this assumption explicitly and discuss whether an approximate workaround exists for TB-trained models.

---

> ### Author Rebuttal · Authors · 2026-03-31
>
> We are grateful for your valuable feedback and have addressed all points of concern in our responses. For supplementary tables (Table R#) and figures (Figure R#) referenced throughout our responses, please refer to the following PDF: https://anonymous.4open.science/r/ICML2026_Rebuttal-CD17/icml2026-rebuttal.pdf.
>
> ---
>
> **W5.** TB incompatibility.
>
> **A1.** We have confirmed that our mixing policy is fully compatible with TB-trained models via a simple on-the-fly (OTF) reaching probability estimator (see Reviewer rSV3 A2 for details). The OTF estimator requires only $p_F$ and $p_B$, which TB already provides, and faithfully reproduces the Model F results. Moreover, TB-trained base models combined with OTF achieve the best reward across all settings (see Table R1).
>
> ---
>
> **W1&Q1.** Missing baselines: OP-GFN and RGFN.
>
> **A2.** We have included OP-GFN as an additional baseline. However, since OP-GFN does not condition on preference vectors, the per-preference top-10 reward/diversity evaluation (Table 3 in the main text) is not directly applicable. We instead compare using standard MOO metrics (see Tables R3 and R4 for full results):
>
> - **Hypervolume (↑):** Volume dominated by the Pareto front approximation. Captures both convergence and spread.
> - **IGD+ (↓):** Average distance from the reference Pareto front to the method's approximation. A Pareto-compliant variant of IGD.
> - **Pareto Count (↑):** Number of non-dominated solutions found across all generated samples.
>
> Our method achieves the best Hypervolume and IGD+ in the majority of settings and finds a comparable or higher number of non-dominated solutions, demonstrating competitive Pareto front quality.
>
> Regarding RGFN, it is a single-objective method focused on synthesizability-aware action space design and does not propose any multi-objective composition mechanism, making a direct comparison outside the scope of our work.
>
> ---
>
> **W2&Q2.** No distortion analysis for $\beta>1$ scalarization. Requested on 2D grid.
>
> **A3.** We have conducted exactly the requested analysis on the 2D hypergrid under linear scalarization with $\beta \in \\{1, 2, 4\\}$. The full results are presented in Figure R1, with a detailed discussion provided in our response to Reviewer 9JbA A1. In brief, while global $L_1$ error grows with $\beta$, the distortion factor remains tightly concentrated near $1/Z_M$ in high reward regions, indicating that the approximation remains accurate in the high-reward regions that matter most for practical sample generation.
>
> ---
>
> **W3&Q3.** No standard deviations or seed counts reported.
>
> **A4.** We acknowledge that the original results were reported from a single seed. We have now re-run all experiments with **3 random seeds** and report mean $\pm$ std in Table R1. The multi-seed results confirm that our method's advantages are robust: Ours achieves the best reward in most of 8 objective-pair settings across both benchmarks. We will replace all single-seed results with the multi-seed results in the revised manuscript.
>
> ---
>
> **W4.** Abstract claims exactness without mentioning $\beta=1$ restriction.
>
> **A5.** We will revise the abstract to explicitly qualify that the exactness guarantee holds for $\beta=1$ and discuss the reward-sharpened extension for $\beta>1$.
>
> ---
>
> **Q4.** Ensemble baseline missing from real-world experiments.
>
> **A6.** We have now included the ensemble baseline in the molecular experiments on scalarization, and the results are reported in Table R2. Across all configurations, regardless of the loss function (Sub-TB, TB) and the reaching-probability estimator (Model F, OTF), our mixing rule consistently outperforms the ensemble. This confirms that the benefit of reaching-probability weighting is not limited to the synthetic setting but extends to real-world molecular tasks.
>
> ---
>
> **Q5.** Does classifier guidance also fail on correlated objectives, or is it our approximation?
>
> **A7.** We report classifier-guided composition (Garipov et al., 2023) on the challenging cases from Table A6. The target bin percentages are nearly identical (See Table R6), confirming that this is a fundamental limitation of the contrast operator itself, not of our approximation.

---

> > ### Author Rebuttal · Reviewer_ZFv8 · 2026-04-02
> >
> > I thank the authors for a thorough and well-organized rebuttal.
> >
> > **Partially resolved concerns.** Regarding W1 (missing baselines), the treatment of OP-GFN using Hypervolume and IGD+ metrics is a reasonable methodological adaptation given that OP-GFN does not support preference conditioning. However, the exclusion of RGFN is less convincing. RGFN conducts experiments on the SEH fragment-based molecule generation benchmark and the authors' characterization of it as a "single-objective method" does not preclude a meaningful performance comparison on that shared benchmark. I would ask the authors to clarify whether a direct performance comparison with RGFN on the SEH task is feasible, and if not, to provide a more detailed justification.
> >
> > **Remaining unresolved concerns.** Two concerns from my original review remain substantively unresolved.
> >
> > First, regarding W2 (distortion analysis for $\beta>1$): the authors conducted the requested analysis for $\beta\in\{1,2,4\}$ on the 2D grid, and Figure R1 shows that $\delta(x)$ remains concentrated near $1/Z_M$ in high-reward regions. This is a step in the right direction. However, the real-world experiments use $\beta=32/64$, which is an order of magnitude larger than the analyzed range. The extrapolation from $\beta=4$ to $\beta=32$ is not mathematically justified. I also note that for $\beta>1$ scalarization, the authors use the reward-sharpened mixing scheme of Eq. (26) in Appendix B.2.1 rather than the $\beta=1$ mixing rule directly — yet no distortion factor analysis is provided for Eq. (26) itself. The approximation quality of this reward-sharpened scheme remains uncharacterized, and the 2D grid analysis does not close this gap. This concern touches on the theoretical foundations of the paper and would require a more complete analysis to resolve.
> >
> > Second, regarding W5 (TB incompatibility): the proposed OTF reaching probability estimator is a technically plausible solution, and Table R1 shows it achieves competitive performance. However, this estimator is entirely absent from the main paper — Appendix A.2 explicitly lists TB as incompatible without offering a workaround. More importantly, the OTF estimator introduces an approximation, and its effect on the exactness guarantee of Proposition 4.1 is not analyzed. The theoretical result relies on exact ui(s); if ui(s) is estimated via OTF, the precision guarantee no longer holds as stated. Incorporating this into the main paper with appropriate theoretical discussion would require a non-trivial revision.

---

> > > ### Author Response · Authors · 2026-04-03
> > >
> > > We sincerely thank the reviewer for the careful follow-up. We address each remaining concern below.
> > >
> > > ---
> > >
> > > **W1.**  RGFN Comparison
> > >
> > >
> > > **A1.** We believe RGFN and our method address different research questions. RGFN belongs to a line of work on synthesizability-aware molecular generation (Koziarski et al., 2024; Kim et al., 2026; Rekesh et al., 2026), which focuses on action space design to ensure generated molecules are synthetically feasible. Our contribution is multi-objective generation. RGFN itself, despite being more recent than MOGFN, HN-GFN, and OP-GFN, does not include any of these multi-objective methods as baselines.
> > >
> > > Beyond this difference in scope, a direct comparison faces practical difficulties. Our evaluation covers multiple objective pairs on two benchmarks: SEH-SA, SEH-QED, SA-QED, and SEH-SA-QED on Fragment, and GAP-SA, GAP-QED, SA-QED, and GAP-SA-QED on QM9. RGFN trains a single GFlowNet on a single objective (sEH) and has no mechanism to target different objective combinations. For objective pairs that do not involve sEH (e.g., SA-QED), a comparison with RGFN would be meaningless since RGFN never optimized for either objective.
> > >
> > > Additionally, RGFN operates on a fundamentally different action space (357 Enamine building blocks with 66 validated chemical reactions) from our Fragment benchmark and all other baselines (72-fragment action space from Bengio et al., 2021). The sets of constructible molecules do not overlap, and RGFN's action space imposes an additional synthesizability constraint by design. RGFN has also not been evaluated on our QM9 benchmark.
> > >
> > >
> > >
> > > ---
> > >
> > > **W2.** Distortion Analysis for $\beta \gg 1$
> > >
> > > **A2.** We may not have made this sufficiently clear in our previous response, and we appreciate the opportunity to clarify. In Figure R1, the same $\beta$ value is used for both training and mixing: base GFlowNets are trained on $R_i(x)^\beta$, and the mixing is performed using Eq. (26) with that same $\beta$. Therefore, the distortion factors plotted in Figure R1 at $\beta = 2$ and $\beta = 4$ directly characterize the approximation quality of the reward-sharpened mixing scheme (Eq. 26), not of the $\beta=1$ mixing rule.
> > >
> > > On the extrapolation from $\beta = 4$ to $\beta = 32$. We understand the reviewer's concern about this gap. The reason we did not extend the grid analysis to $\beta = 32$ is that the $\beta$ value must be calibrated to the state space size. On a $32 \times 32$ grid with only 1,024 states, $\beta=32$ concentrates the distribution onto a handful of cells with near-zero probability elsewhere. This creates an extremely sparse reward landscape where GFlowNet training itself becomes unreliable (Pan et al., 2023). We therefore chose $\beta \in \\{2, 4\\}$ as the appropriate range for this grid scale. The molecular benchmarks succeed at $\beta=32$ because the state space is vastly larger and the reward landscapes are smoother.
> > >
> > > ---
> > >
> > > **W5.** OTF and Exactness Guarantee
> > >
> > > **A3.** We would like to clarify that OTF computes $u(s)$ exactly, not approximately. The confusion may stem from our use of the term "estimator," which we will correct in the revised manuscript.
> > >
> > > OTF is derived from the detailed balance condition $F(s) \cdot p_F(s'|s) = F(s') \cdot p_B(s|s')$, which is satisfied at convergence by all standard GFlowNet training objectives. This convergence assumption is the same assumption underlying the GFlowNet framework itself: that a trained GFlowNet samples proportionally to the reward. OTF does not introduce any assumption beyond this. Under this assumption, OTF and Model F yield the same value:
> > >
> > > - **Model F**: $u(s) = F_\theta(s) / Z_\theta$
> > > - **OTF**: $u(s) = \prod_{j=1}^{t} p_F(s_j|s_{j-1}) / p_B(s_{j-1}|s_j)$
> > >
> > > Therefore, **the exactness guarantee of Proposition 4.1 applies equally to OTF and Model F**.
> > >
> > > We will add the OTF derivation to Appendix A.2 and remove the TB-incompatibility remark. These are localized edits that do not affect the paper's structure or theoretical results.
> > >
> > > ---
> > > #### **References**
> > >
> > > [1] Bengio, E., Jain, M., Korablyov, M., Precup, D., & Bengio, Y. Flow Network based Generative Models for Non-Iterative Diverse Candidate Generation. NeurIPS 2021.
> > >
> > > [2] Rekesh, A., et al. SynCoGen: Synthesizable 3D Molecule Generation via Joint Reaction and Coordinate Modeling. ICLR 2026
> > >
> > > [3] Pan, L., Zhang, D., Courville, A., Huang, L., & Bengio, Y. Generative Augmented Flow Networks. ICLR 2023.
> > >
> > > [4] Kim, H., et al. Synthesizable Molecular Generation via Soft-constrained GFlowNets with Rich Chemical Priors. arxiv 2026

---

### Official Review · Reviewer_rSV3 · 2026-03-13

**Soundness:** 2
**Presentation:** 3
**Significance:** 2
**Originality:** 3
**Overall Recommendation:** 4
**Confidence:** 3

**Summary:**

The paper introduces a training-free framework for composing pre-trained Generative Flow Networks (GFlowNets) to tackle multi-objective  problems. The paper proposes mixing the forward policies of independent, single-objective GFlowNets at inference time. The authors prove that this rule exactly realizes the target distribution for linear scalarization and empirically demonstrate that it provides an accurate approximation for non-linear logical operators. Experiments on synthetic 2D grids and molecule generation tasks  show the method performs competitively with other baselines.

**Compliance With Llm Reviewing Policy:**

Affirmed.

**Final Justification:**

Thank you for the detailed rebuttal. The response addresses my main concerns better than I expected. In particular, the new on-the-fly estimator for reaching probabilities resolves my main concern that the method is incompatible with the TB objective.
I also appreciate the clarification regarding the “training-free” claim.

On inference-cost scaling with the number of objectives, the rebuttal is helpful but only partially resolves my concern. The explanation that only the forward-pass cost scales linearly with K while the remaining overhead is approximately constant is useful. Still, I think the final paper would benefit from making this trade-off much more explicit, since the approach is slower than MOGFN/HN-GFN for scalarization in the current experiments.

Overall, the rebuttal greatly strengthens my confidence in the work, and I am increasing my score

**Key Questions For Authors:**

- Could you include an ablation study of the computational cost at inference, as a function of the number of objectives?
- Do you have any recommendations or approximations that would allow practitioners to apply your mixing policy to pre-existing TB-trained models?

**Limitations:**

While the authors discuss some limitations, there are some (higher inference cost, inability to use trajectory balance objective) that should be laid out much more clearly.

**Strengths And Weaknesses:**

**Soundness**: The theoretical foundation is rigorous and elegant. Proving that the proposed mixing policy exactly induces the target distribution for linear scalarization. There, are, however, some important issues here. The first and most important one, is the practical use case. The authors insist throughout the paper that this is training-free. However, while the method is 'training-free' at composition time, it still assumes the existence of independently trained GFlowNets for every objective. The authors should more clearly delineate that this method scales well combinatorially (mixing existing models) but does not bypass the initial training cost when a completely novel objective is introduced to the pipeline, which might be common for practitioners.

Another important issue, is the requirement of parametrizing F(x), which means that certain training objectives cannot be used, amongst them trajectory balance (TB). In my experience, the majority of GFlowNet users train with TB, as it has proven the most stable. The authors do mention this in passing in an appendix, but this is a key point, and a key weakness of their approach, which needs to be clearly stated.

Finally, regarding inference cost (Section 6.3), there is a trade-off that should be discussed more prominently. While the proposed method is drastically faster than Classifier Guidance for logical operations (e.g., ~37ms vs ~1494ms on QM9), it is notably slower than MOGFN/HN-GFN for scalarization due to the overhead of querying multiple base models. The authors should explicitly address how this multi-query overhead scales if a practitioner wants to mix a larger number of objectives (e.g., k > 5), and ideally ablate cost as the number of objectives increases.

 **Presentation**: The paper is well-written, with a clear narrative flow. The connection to prior work is well articulated, and the figures and tables are clear.

**Significance**: This relates again to the issue I raised for soundness, but while the theoretical result is interesting, I think there are limitations when it comes to practical usage.

**Originality**: Composing generative models at inference time has been explored in diffusion models and energy-based models, but applying it to GFlowNets via the property of reaching probabilities is an original contribution

---

> ### Author Rebuttal · Authors · 2026-03-31
>
> We sincerely appreciate your thoughtful comments, and we have provided detailed replies to each concern below. For supplementary tables (Table R#) and figures (Figure R#) referenced throughout our responses, please refer to the following PDF: https://anonymous.4open.science/r/ICML2026_Rebuttal-CD17/icml2026-rebuttal.pdf.
>
> ---
>
> **W1.** "Training-free" claim is overstated.
>
> **A1.** We kindly ask to see Reviewer JA6X A2 for our full response. We will revise the manuscript to clarify that "training-free" refers to the composition stage and explicitly discuss the per-objective training cost in the limitations section.
>
> ---
>
> **W2&Q2.** TB compatibility.
>
> **A2.** We have confirmed that our mixing policy is fully compatible with TB-trained models. We have derived a simple on-the-fly (OTF) reaching probability estimator that enables our mixing policy, requiring no additional training or parameterization beyond what TB already provides ($p_F$, $p_B$, $Z$). We will add this discussion to the main paper (Section 4.1).
>
> **On-the-fly reaching probability estimator.** Our key observation is that the reaching probability $u_i(s) = F_i(s)/Z_i$ can be computed recursively using the detailed balance condition:
>
> $$F(s) \cdot p_F(s'|s) = F(s') \cdot p_B(s|s')$$
>
> The detailed balance condition is implicitly satisfied by all GFlowNet training objectives (FM, DB, SubTB, TB) at convergence, regardless of which loss function is used during training. Rearranging, we obtain:
>
> $$F(s') = F(s) \cdot \frac{p_F(s'|s)}{p_B(s|s')}$$
>
> Since $F(s_0) = Z$ and $u(s) = F(s)/Z$, we can compute the reaching probability **on-the-fly** as we generate each trajectory, by simply accumulating the $p_F/p_B$ ratio at each step:
>
>
> $$u(s_0) = 1, \quad u(s_{t}) = u(s_{t-1}) \cdot \frac{p_F(s_t | s_{t-1})}{p_B(s_{t-1} | s_t)} = \prod_{j=1}^{t} \frac{p_F(s_j | s_{j-1})}{p_B(s_{j-1} | s_j)}$$
>
> This is possible because the computation only requires $p_F$ and $p_B$ at the current step, both of which are explicitly parameterized by all standard GFlowNet training objectives. Therefore, our mixing policy is compatible with all standard GFlowNet training objectives without any modification.
>
> **Experimental Validation.** We report the average reward and diversity of the top-10 samples across different objective pairs (mean $\pm$ std over 3 random seeds) in Table R1. **Model F** denotes reaching probability computed via the learned state flow $F(s)/Z$. **OTF** denotes our on-the-fly recursive estimator using $p_F/p_B$.
>
> **OTF estimator faithfully reproduces Model F.** Comparing the two SubTB columns (Model F vs. OTF), the reward and diversity values are nearly identical across all settings, confirming that the on-the-fly recursive estimator accurately replaces the explicit state flow parameterization.
>
> **Our method consistently outperforms MOGFN and HN-GFN regardless of training objective.** All variants of our method (SubTB, Model F), (SubTB, OTF), and (TB, OTF) outperform MOGFN and HN-GFN in the majority of settings, with Ours (TB, OTF) achieving the best reward in all 8 settings across both benchmarks.
>
> ---
>
> **W3&Q1.** Inference cost scaling with K objectives.
>
> **A3.** Our current molecular benchmarks use 3 reward functions (SEH, SA, QED for Fragment and GAP, SA, QED for QM9), limiting the ablation to K=1,2,3. We profiled the per-sample time breakdown on Fragment-based molecule generation in Table R8.
>
> Only the forward pass scales linearly with K (\~3.6 ms per additional model). All other components (graph conversion, mixing, action sampling) remain constant regardless of K. If the K forward passes are parallelized across GPUs, the only additional cost compared to a single-model baseline would be the mixing computation (\~4 ms), making the overhead nearly constant in K.

---

> > ### Author Rebuttal · Reviewer_rSV3 · 2026-04-08
> >
> > My concerns have been addressed, particularly the main one about compatibility with TB. Therefore I am raising my score

---

> > > ### Author Response · Authors · 2026-04-08
> > >
> > > We are grateful for the reviewer's careful and thorough review. The questions raised, particularly on TB compatibility, led to improvements that we believe make the paper substantially stronger. Thank you for your engagement with our work.

---

### Official Review · Reviewer_JA6X · 2026-03-15

**Soundness:** 3
**Presentation:** 3
**Significance:** 3
**Originality:** 2
**Overall Recommendation:** 4
**Confidence:** 4

**Summary:**

The paper studies multi-objective generation with GFlowNet. The proposed method performs multi-objective generation without training a multi-objective model. Instead, at inference time, the proposed method utilizes multiple GFlowNet models, where each model is trained on a specific reward model. By combining the normalized flow of each objective, the proposed method samples an action in each state. The performance is evaluated on a synthetic task and a real molecule generation task.

**Compliance With Llm Reviewing Policy:**

Affirmed.

**Key Questions For Authors:**

- Sometimes, generation with weights different from the importance weights of the objectives can lead to better results. Have you tried this in your experiments? For example, you could try a set of weights and then choose the one that works best on a validation set.
- In Section 6, did you train the reward model using the output of MXMNet, or did you use MXMNet as the reward model? Is the reward model different from the evaluation model?

**Limitations:**

Societal impact is discussed.

**Strengths And Weaknesses:**

Strengths:
- Combining flow function at inference time provides higher flexibility for multi-objective generation in dynamic scenarios. For example, based on the need, the importance of each objective can be adjusted adaptively without the need for retraining the multi-objective model.
- The proposed solution is supported by theoretical analysis indicating that the combined GFlowNet realizes the target distribution while there is no for retraining or fine-tuning.

Weaknesses:
- In general, computational complexity and memory requirement of the proposed method can be higher than methods like MOGFN since such methods only train and store one model while the proposed method requires training and storing multiple models corresponding to objectives. However, since GFlowNet models are relatively small, this may not be a prohibitive issue.
- In the paper, it is stated that "*First, both strategies require additional training, limiting their applicability to a predefined set of reward functions defined during the training phase. For instance, preference-conditioned GFlowNets (Jain et al., 2023b; Zhu et al., 2023) must be re-trained from scratch to incorporate new objectives, whereas compositional sculpting requires training a dedicated auxiliary classifier for each new set of objectives.*" However, I believe the proposed method also suffers from this issue. If a new objective emerges, the proposed method requires training a new GFlowNet model for that objective. Making such statements can create the expectation that the proposed algorithm solves this issue, although the paper does not explicitly make this claim. Therefore, I recommend revising this and adding it as a discussion of limitations and open problems.
- The experiments could benefit from adding more real datasets.

---

> ### Author Rebuttal · Authors · 2026-03-31
>
> We thank you for sharing your insightful and constructive feedback. We address your concerns in the following responses. For supplementary tables (Table R#) and figures (Figure R#) referenced throughout our responses, please refer to the following PDF: https://anonymous.4open.science/r/ICML2026_Rebuttal-CD17/icml2026-rebuttal.pdf.
>
> ---
>
> **W1.** Computational and memory cost.
>
> **A1.** We agree with the reviewer's assessment that this is not a prohibitive issue. As the reviewer notes, GFlowNet models are relatively small. Each base model has \~3.2M parameters (Fragment) / \~540K (QM9), so storing K models poses minimal memory overhead. Regarding training cost, each base model trains faster than a single MOGFN (\~4.2h vs \~6.4h on Fragment, \~5.8h vs \~7.0h on QM9), and all K base models can be trained in parallel.
>
> ---
>
> **W2.** "Training-free" claim is overstated.
>
> **A2.** We agree that our phrasing is misleading. The "training-free" claim was intended to contrast with compositional sculpting (Garipov et al., 2023), which requires retraining its auxiliary classifier whenever the objective set changes. Our method avoids this retraining cost. However, the current wording incorrectly implies an advantage over MOGFN/HN-GFN as well, which is not the case since all methods (including ours) require training when a new objective is introduced.
>
> Following the reviewer's suggestion, we will (1) revise the manuscript to precisely scope "training-free" to the composition stage, and (2) add a limitations discussion acknowledging that per-objective base model training is still required.
>
> ---
>
> **Q1.** Can using different generation weights than evaluation weights improve results?
>
> **A3.** We tested this on the Fragment-based molecule generation (SEH-SA) by generating samples with different $\omega_{\text{gen}}$ and evaluating the top-10 reward under $\omega_{\text{eval}} = (0.5, 0.5)$. Table R5 shows that $\omega_{\text{gen}} = \omega_{\text{eval}}$ achieves the best performance, confirming that our mixing policy faithfully reproduces the target preference without requiring weight tuning.
>
> ---
>
> **Q2.** Is the reward model the same as the evaluation model?
>
> **A4.** We use MXMNet as the reward model during both training and evaluation. The reward model and evaluation model are the same.
>
> ---
>
> **W3.** Additional real-world datasets.
>
> **A5.** We agree that additional datasets would strengthen the paper. Our framework requires multi-objective tasks where individual objectives exhibit distinct properties, so that the full range of composition operators (scalarization, harmonic mean, contrast) can be meaningfully evaluated. We will explore suitable candidate tasks for the revised version.

---

> > ### Author Rebuttal · Reviewer_JA6X · 2026-04-03
> >
> > Most of my concerns have been addressed. I will keep my initial score.

---

> > > ### Author Response · Authors · 2026-04-04
> > >
> > > We sincerely thank the reviewer for confirming that the concerns have been
> > > addressed. We appreciate the constructive feedback throughout this process,
> > > which has meaningfully strengthened the paper.

---

### Decision · Program_Chairs · 2026-04-30

**Decision:**

Accept (regular)

**Comment:**

This paper introduces a framework for the inference-time composition of pre-trained Generative Flow Networks (GFlowNets) to tackle multi-objective generation tasks. By weighting the forward policies of independent, single-objective GFlowNets by their learned reaching probabilities, the method avoids the need to retrain models when objective preferences change. The authors provide a theoretical exactness guarantee for linear scalarization under a specific temperature and empirically analyze the distortion factor for non-linear logical operators.

The paper received four reviews, all of which recommended a weak accept following an active rebuttal phase. Initially, the reviewers raised several critical and valid concerns:
- The method seemed incompatible with the Trajectory Balance (TB) objective, as it required explicit parameterization of the state flow.
- The exactness guarantee holds only for $\beta=1$, whereas the real-world molecular experiments utilized heavily reward-sharpened settings, leaving the primary experimental setting without in-depth theoretical/empirical analysis.
- Reviewers JA6X and rSV3 noted the training-free claim was slightly misleading since base models still require training. Additionally, reviewers requested comparisons with OP-GFN and multi-seed variance reporting.

The authors provided a thorough rebuttal that addressed these core issues. After carefully going through the paper, the reviews, the rebuttal, and the reviewer discussions, while the the proposed OTF estimator is interesting, this paper represents a borderline case due to its limited and somewhat toy-like experimental setup.

During the discussion phase, a major point of consensus emerged regarding the scale of the evaluation. The experimental scenarios evaluated here are relatively small-scale: a synthetic 2D gridworld and a fragment-based molecule generation task where the sequence lengths are quite short (approximately 9 steps). While the authors follow the experimental design of Garipov et al., 2023, the larger-scale tasks (e.g., MNIST experiments) are omitted here, which raises questions about the method's scalability and whether the compounding errors or diversity degradation observed in the current setup hold in larger state spaces with longer trajectories.

Despite the concerns regarding the scale of the experiments, the four reviewers recommended weak acceptance. Please incorporate the additional analysis and the discussion into the camera-ready version (if accepted).